# Folding and unfolding: A topological framework for understanding intangible cultural heritage tourism in urban villages - The case of chebei dragon boat scenery, Guangzhou, China

**Xixi Tang, Shengchao Li**  *

School of Management, Guangdong University of Education, Guangzhou, China

* lishengchao@qq.com

## Abstract

### Background

Rapid urbanization poses unprecedented challenges to intangible cultural heritage (ICH) preservation, particularly within urban villages where traditional practices face displacement pressures. While heritage tourism offers potential solutions, existing frameworks inadequately address how ICH spaces maintain cultural authenticity while undergoing radical transformation.

### Methods

This study introduces a novel topological framework to analyze ICH space dynamics, conceptualizing them as entities capable of continuous deformation while preserving fundamental properties. Using grounded theory methodology, we conducted 38 in-depth interviews, extensive participant observation, and comprehensive document analysis of the Chebei Dragon Boat Scenery in Guangzhou, China.

### Results

Our analysis reveals a "2-6-18" generative logic underlying ICH space structure and reproduction. We identify "cultural experience" as the core topological invariant that enables cultural continuity despite spatial transformation. ICH spaces are reproduced through dynamic interactions within a "culture-power-capital" three-dimensional matrix, facilitated by tourism-driven "unfolding" processes that make implicit cultural knowledge explicit while maintaining essential properties.

### Conclusions

The findings demonstrate how "urban renewal + cultural tourism integration" can serve as an effective spatial reproduction mechanism for ICH preservation. This topological approach offers a robust analytical framework for heritage tourism management, moving

**Data availability statement:** Data cannot be shared publicly because of ethical restrictions. The data underlying this study consists of transcripts from 38 in-depth interviews and field notes containing personal information and detailed narratives about local community dynamics. Although the data has been anonymized, the detailed nature of qualitative descriptions could potentially lead to the re-identification of participants within the specific village context. The Ethics Committee of Guangdong University of Education imposed these restrictions to protect participant privacy and confidentiality, as per the approved ethical protocol (Ref: 20251028). However, we have provided the Comprehensive Interview Guide (S1 File) and the Detailed Grounded Theory Coding Structure (S2 File) as Supporting Information, which constitute the minimal data set necessary to interpret the findings. Researchers who meet the criteria for access to confidential data may request access to the anonymized transcripts by contacting the Academic Committee of Guangdong University of Education at: lishch3@mail2.sysu.edu.cn.

**Funding:** This research was funded by the Philosophy and Social Sciences Planning Office of Guangdong Province (Grant No. GD23YGL12 to XT and GD24YGL29 to SL); Department of Education of Guangdong Province (Grant No. 2024WTSCX158 to SL and 2025WTSCX082 to XT); and Bureau of Education of Guangzhou Municipality (Grant No. 2024BQWGC008). The funders had no role in the study design, data collection, analysis, and decision to publish, except for the Publication Fee.

**Competing interests:** The authors have declared that no competing interests exist.

beyond binary preservation-development paradigms toward adaptive sustainability models that honor both cultural continuity and contemporary urban development needs.

## Introduction

The ancient Chinese saying "northerners excel at archery, southerners are skilled with boats" reflects the profound connection between regional culture and geographic conditions. During the 2024 Dragon Boat Festival period, Guangzhou's dragon boat culture, combined with the distinctive atmosphere of urban villages filled with traditional embroidered umbrellas, dragon boat racing, communal dragon boat rice, firecrackers, gongs and drums, and crowds of landlords, tenants, and tourists, created an exceptionally vibrant folk festival celebration. Guangdong Province's cultural and tourism industry capitalized on this popularity, prominently featuring "Dragon Boat Festival" labels across various promotional platforms.

In the water villages of the Pearl River Delta within the Greater Bay Area, dragon boat culture has witnessed the growth and reproduction of Cantonese people for over a millennium. However, the continuous impacts of rapid urbanization have caused numerous traditional villages to become urban "depressions"—spaces known as urban villages—where dragon boat traditions and many other millennium-old folk customs face various difficulties under modern urban encirclement, surviving precariously in overcrowded urban village spaces and gradually declining [1].

The transformation of original lifestyles and production methods triggers chain reactions including the decline of clan society, changes in family structure, transformation of livelihood methods, and large-scale population mobility [2]. These changes lead to the disappearance of application scenarios where intangible cultural heritage (ICH) existed, causing excellent traditional culture embedded within ICH to face extinction crises. Feng Jicai argues that "traditional villages store extremely rich intangible spiritual cultural heritage, inheriting the Chinese nation's historical memory, production and life wisdom, cultural and artistic crystallization, and ethnic regional characteristics" [3].

Recently, people's nostalgia for original ecological and traditional rural culture has gradually become fashionable, especially with the rise of grassroots-level cultural tourism. Cultural tourism projects featuring low consumption barriers, accessibility, warmth, and cultural sentiment have been welcomed by new generations. Exploring ICH spaces within urban villages and investigating their integration with urban village renewal represents both a natural progression and a strategic opportunity [4].

To enable ICH to burst with new vitality under the new normal of urban renewal and high-quality development, and to promote proactive urban development in urban villages, it is essential to comprehend the essence and value of ICH spaces in urban villages. This requires addressing fundamental questions: What is the relationship between ICH spaces in urban villages and urban spaces? What constitutes the evolutionary survival logic of ICH spaces within urban village environments? How can ICH spaces in urban villages adapt to contemporary social changes to achieve reproduction?

## Literature review and theoretical framework integration

**The challenge of urban village transformation and cultural heritage.** Urban villages represent crucial spatial types within Chinese cities [5]. As pain points in coordinated urban spatial development [6], different cities have conducted extensive urban village renewal explorations based on regional characteristics, governance needs, and development trends, often converging on renewal models dominated by real estate development [7]. This singular emphasis on economic goal-oriented transformation contradicts the original intention of solving urban problems and empowering cities with new vitality through urban renewal [8,9].

The evolution of Western urban renewal over more than half a century provides the insight that successful urban renewal requires attention to cultural heritage protection and emphasizes culture-oriented urban renewal models [10,11]. Urban village renewal constitutes the process of absorption by rapidly expanding cities, involving not merely economic transformation, institutional change, industrial deployment, and organizational transformation, but also cultural protection, local identity, emotional attachment, and other dimensions [12]. This presents a prolonged and complex process encompassing spatial institutional transformation, economic transformation, and cultural transformation [13,14].

Current research on urban village transformation predominantly focuses on stakeholder perspectives to evaluate renewal effectiveness [15] and explores environmental improvements through specific infrastructure and public services [16]. While these approaches provide valuable insights, they often position urban villages as passive subjects of transformation rather than recognizing their potential for proactive adjustment and trend-following renewal models. Furthermore, studies concentrating on tangible physical spaces that can be directly perceived are numerous [17], yet substantial exploration space remains for intangible spatial transformation involving social behavior, spatial relationships, discourse systems, conceptual order, and sustainable spatial governance.

**Intangible cultural heritage: Conceptual evolution and spatial dimensions.** The trajectory from UNESCO's "Convention for the Safeguarding of the Intangible Cultural Heritage" to China's "Intangible Cultural Heritage Law," alongside various scholarly investigations of ICH concepts, connotational evolution, values, and protection systems [18,19], demonstrates that under diverse social backgrounds, ICH's multidimensional conceptual system remains fundamentally stable. This system primarily encompasses traditional folk culture and its cultural spaces [20,21].

Within the context of global economic integration and cultural convergence trends, this culture and cultural space face particular pressures. The emergence of grassroots cultural tourism, characterized by low consumption barriers, accessibility, warmth, and cultural sentiment appealing to new generations, creates both opportunities and challenges for ICH preservation [22]. Exploring ICH spaces within urban villages and investigating their integration with urban village renewal represents both timely recognition of current trends and strategic positioning for future development.

**Theoretical bridging: Topology as analytical framework for cultural dynamics.** The application of topological thinking to cultural heritage studies addresses fundamental theoretical limitations in existing frameworks (**Fig 1**). Contemporary heritage studies have formed a core consensus that the essence of ICH is dynamic, its survival dependent on continuous evolution and adaptation in transmission and reproduction [23–25]. This consensus is based on the core principles of "living heritage" and "community participation" emphasized in UNESCO's 2003 Convention. "Cultural space" this study focuses more on its dynamic properties as a social process a field continuously shaped by cultural practice, social relations, and power dynamics. The "topological turn" in social sciences provides a rich theoretical resource using topology as a powerful "heuristic metaphor" and "analytical language" to describe the structural resilience of ICH space rather than pursuing strict mathematical formalization [23–25].

Topology, as a mathematical discipline focusing on properties preserved under continuous deformation [26], provides powerful analytical tools for understanding cultural persistence within spatial transformation. Unlike Euclidean geometry's emphasis on fixed measurements and shapes, topology examines relational properties that endure despite surface-level transformations. This mathematical concept proves particularly relevant for cultural heritage analysis because it enables

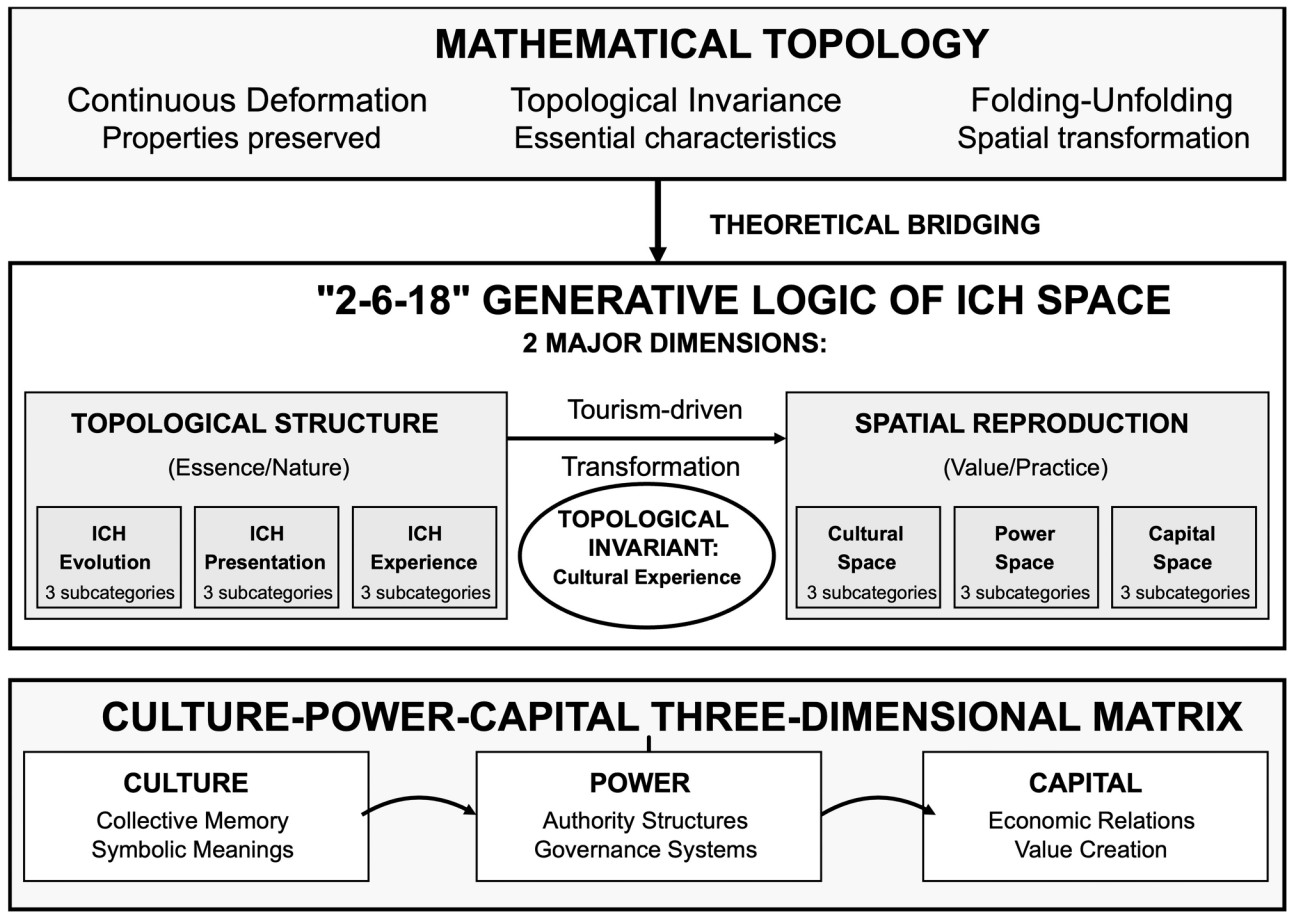

**Fig 1. Topological framework for ICH space analysis.**

identification of invariant properties within ICH practices while their physical manifestations and social contexts undergo significant change.

The theoretical bridging between topology and cultural studies operates through several key mechanisms. First, topological invariance provides a conceptual framework for understanding cultural continuity that transcends material fixity. It is important to emphasize that the "topological invariant" used here is a metaphorical borrowing from mathematics, not a strict conceptual equivalence. In this socio-cultural context, it refers to the core element that maintains a stable, connective function throughout dynamic transformation—which this paper identifies as "cultural experience." This "invariant" is the relational kernel and driving force that persists, capable of connecting different times, spaces, and social dimensions. Cultural practices maintain their essential characteristics not through unchanging physical forms but through preserved relational structures and experiential qualities. Second, the topological concepts of "folding" and "unfolding" offer metaphorical tools for analyzing how cultural meaning becomes compressed within contemporary spaces during transformation (folding) and subsequently made explicit through tourism and development processes (unfolding).

Every locality or region can be understood as space suitable for certain specific social activities and behaviors, with these activities and behaviors constituting its cultural core. Different spaces become shaped by core culture into various distinctive types [27]. ICH spaces in urban villages function as ritualized memorial local activity venues, representing traces of repeated customs accumulated in temporal and spatial dimensions over centuries. These spaces can be

conceptualized as temporal folding connection points where past time regions connect with the present through folding mechanisms, with this connection point representing the core culture.

The temporal-spatial associations of ICH spaces in urban villages align remarkably with topology's concept of topological invariance. Folding and connection points constitute specialized terminology within topology [28]. Topology focuses on properties that objects can maintain unchanged after continuous deformation. From a topological perspective, topological folding connects distant with proximate, exceptional with normative, maintaining strong regional cultural continuity and inheritance.

**Spatial production theory and the culture-power-capital matrix.** To comprehensively understand how ICH spaces are produced and reproduced within urban villages, particularly under tourism influence, this study integrates spatial production theory while extending Henri Lefebvre's foundational insight that space is not passive backdrop but actively produced through social relations, power dynamics, and economic forces [29]. We propose a "culture-power-capital" three-dimensional matrix as the analytical framework for understanding ICH space dynamics. This three-dimensional framework resonates profoundly with sociologist Pierre Bourdieu's theories of field, capital, and habitus [30]. Bourdieu's concepts, especially cultural and symbolic capital, provide the core lens for the "Capital Space" dimension [31]. Specifically: The "Cultural Space" dimension, while drawing from UNESCO, also inherits Henri Lefebvre's theory of "the production of space," treating space as a living social product [29]. The "Power Space" dimension draws from Michel Foucault's discussions on how power operates and is objectified through spatial arrangements [32]. The "Capital Space" dimension, as noted, primarily borrows from Pierre Bourdieu's capital theory [30].

This tripartite framework operates across multiple scales and temporal dimensions. Cultural space encompasses collective memory, symbolic meanings, identity narratives, and practiced traditions that define community heritage. Within urban villages, cultural space exhibits topological properties through its capacity for folding temporal layers, creating dense nodes of meaning and memory. Power space refers to configurations of authority, control, and decision-making governing ICH spaces, including formal government regulations, traditional leadership structures, and emergent stakeholder networks. Capital space encompasses economic relations, property regimes, and value creation processes associated with ICH, incorporating both traditional economies and increasingly prominent tourism revenues.

The dynamic interaction among these dimensions produces "ICH space assemblages"—complex configurations where cultural meaning, social power, and economic value interact in mutually constitutive ways [33]. Tourism development reconfigures these assemblages by introducing new actors, values, and practices, potentially maintaining topological properties through careful management or disrupting them through inappropriate intervention [34,35].

It must be pointed out that any theoretical model has its focus and boundaries. The topological framework's advantage lies in revealing the internal logic and dynamic structure of cultural meaning reproduction, making it suitable for understanding adaptation and continuity under management intervention. For deeper causal mechanisms of institutional contradictions or the excessive commercial exploitation of cultural symbols by capital—while the "power" and "capital" dimensions bring them into view—a more specialized theoretical toolkit, such as institutional analysis or political economy, would be required for a full investigation.

## Research context and scale considerations

We examine these dynamics through an in-depth case study of the Chebei Dragon Boat Scenery in Guangzhou, China. The selection of this case enables multi-scalar analysis spanning individual, community, urban, and regional levels. Chebei Village, a millennium-old settlement now embedded within one of China's largest metropolises, provides an ideal microcosm for investigating ICH persistence within urban transformation while connecting to broader regional cultural networks.

Despite radical physical and demographic changes—the village currently houses over 60,000 residents, 80% of whom are migrants—Chebei maintains a vibrant dragon boat tradition that attracts hundreds of boats and over 100,000

spectators annually [36,37]. This enduring cultural vitality, coupled with recent heritage tourism development initiatives and the village's unique collective economy based on ancestral land rental income, offers rich empirical grounds for exploring ICH space dynamics across scales.

Our research addresses two interrelated questions operating at different analytical scales: How do ICH spaces within urban villages maintain their essential cultural properties while undergoing significant spatial and socio-economic transformation at local, community, and regional levels? What mechanisms, particularly through heritage tourism, facilitate the adaptive reproduction of these ICH spaces across temporal and spatial scales?

## Methods

### Research design and philosophical approach

This study employed grounded theory methodology, a systematic approach for developing theoretical understanding from empirical data particularly suited for exploring complex, under-theorized phenomena like ICH space dynamics in rapidly changing urban contexts. Grounded theory's emphasis on inductive theory building enables the emergence of topological insights from data rather than imposing predetermined mathematical concepts, ensuring empirical grounding for theoretical innovation.

The research followed Strauss and Corbin's systematic procedures while incorporating recent methodological developments in constructivist grounded theory [38]. Data collection and analysis proceeded iteratively, with preliminary observations guiding theoretical sampling and emerging concepts directing subsequent fieldwork. This iterative relationship between data collection and analysis facilitated progressive theory development, moving from initial observations suggesting themes of spatial transformation toward targeted sampling of participants experiencing different aspects of change across various temporal and spatial scales.

### Research setting: Chebei village and dragon boat heritage

Chebei Village, located in Tianhe District, Guangzhou, serves as our primary case study site (**Fig 2**). Founded during the Tang Dynasty (618–907 AD) and flourishing during the Song Dynasty, Chebei represents one of Guangzhou's oldest and largest urban villages. The village's strategic position within the Pearl River Delta's tributary system has historically made it a natural gathering point for regional dragon boat activities, contributing to its designation as "official scenery" for dragon boat races during the Ming Dynasty.

The village's transformation from an agricultural settlement to a dense urban enclave housing over 60,000 residents exemplifies broader urbanization challenges facing traditional communities across Asia while maintaining distinctive cultural characteristics. Despite radical demographic and physical changes, Chebei uniquely preserves a robust dragon boat tradition dating to the Song Dynasty, demonstrating remarkable cultural resilience through various historical disruptions including the Cultural Revolution before experiencing vigorous revival alongside contemporary urbanization processes.

Contemporary Chebei maintains essential cultural infrastructure that supports ongoing heritage practices while accommodating urban development pressures. The village's cultural landscape encompasses both traditional spaces dedicated to ancestral worship and community governance, as well as newly developed facilities designed to preserve and interpret cultural heritage for broader audiences. This infrastructure provides the material foundation for cultural continuity while enabling adaptive responses to contemporary tourism development opportunities.

The spatial organization of Chebei's cultural heritage reflects the complex layering of historical development and contemporary adaptation. Six key sites exemplify this cultural-spatial integration (**Fig 3**): The Dragon Boat Cultural Exhibition Hall (1) represents innovative heritage interpretation, serving as Guangzhou's first village-level ICH preservation and education facility since its establishment in 2016. This community-initiated museum materializes collective memory through artifact display and narrative construction, demonstrating how villages can assume agency in heritage preservation and tourism development.

 

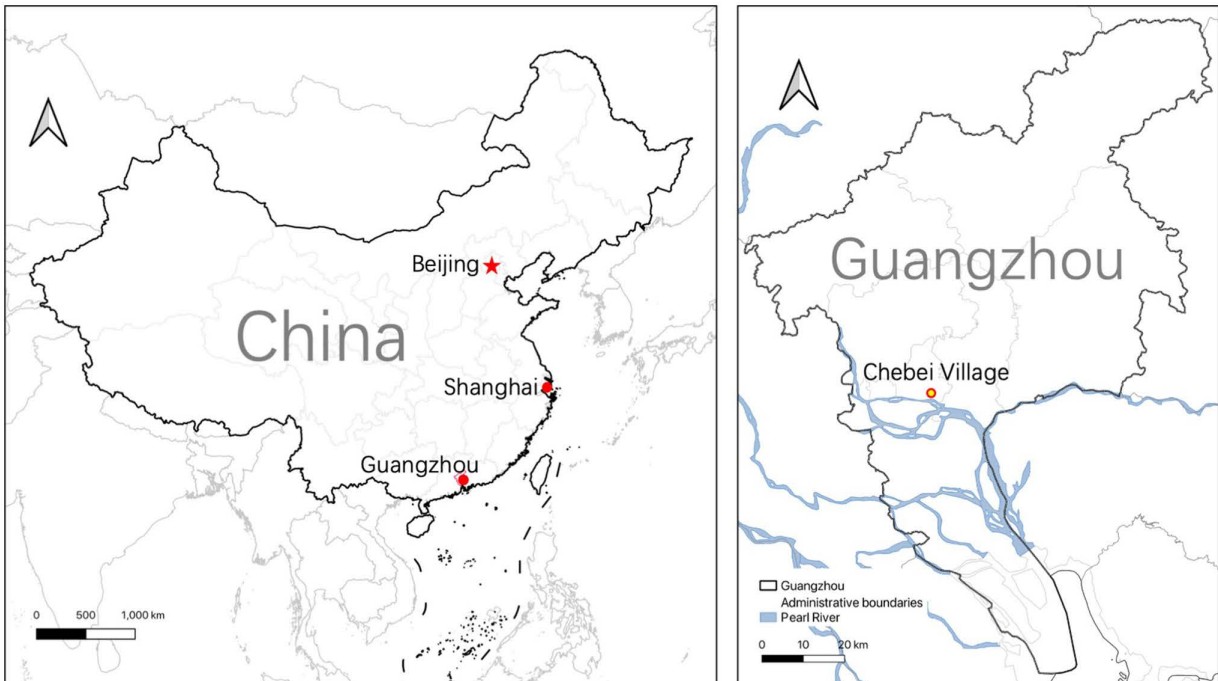

**Fig 2. Location Map of Chebei Village, Guangzhou (This map is drawn with QGIS software and the map data is sourced from Natural Earth (https://www.naturalearthdata.com/)).**

Chebei Creek (2) constitutes the vital waterway enabling dragon boat activities, with its 800-meter straightaway and 50-meter width providing rare urban space capable of accommodating large-scale competitive racing. Environmental restoration investments exceeding 100 million RMB have transformed this waterway from severely polluted industrial drainage to Class III water quality suitable for cultural activities, literally enabling cultural revival through ecological rehabilitation.

The Dragon Boat House (3) functions as both practical storage facility and symbolic community landmark, housing the fleet of racing vessels while maintaining their visibility as cultural markers throughout the year. This facility exemplifies how functional infrastructure can serve cultural purposes, supporting both practical maintenance requirements and symbolic community identity expression.

Traditional ancestral halls, including Qingchuan Su Ancestral Hall (4) and Longxing Su Ancestral Hall (5), maintain their historical roles as centers of clan organization and cultural authority while adapting to contemporary governance requirements. These spaces continue hosting ritual activities, community meetings, and cultural education while serving as focal points for dragon boat association organization and festival coordination. Their preservation within dense urban development demonstrates how traditional authority structures can maintain legitimacy and functionality through adaptive institutional arrangements.

The Xihua Memorial Archway (6) represents the broader cultural landscape connecting Chebei to regional heritage networks, serving as both physical landmark and symbolic marker of historical significance. Such monuments demonstrate how individual communities connect to broader cultural systems while maintaining distinctive local identity, illustrating the multi-scalar nature of heritage preservation and cultural continuity.

This cultural infrastructure operates as an integrated system supporting both traditional community functions and contemporary tourism development. The spatial arrangement enables authentic cultural practice while providing

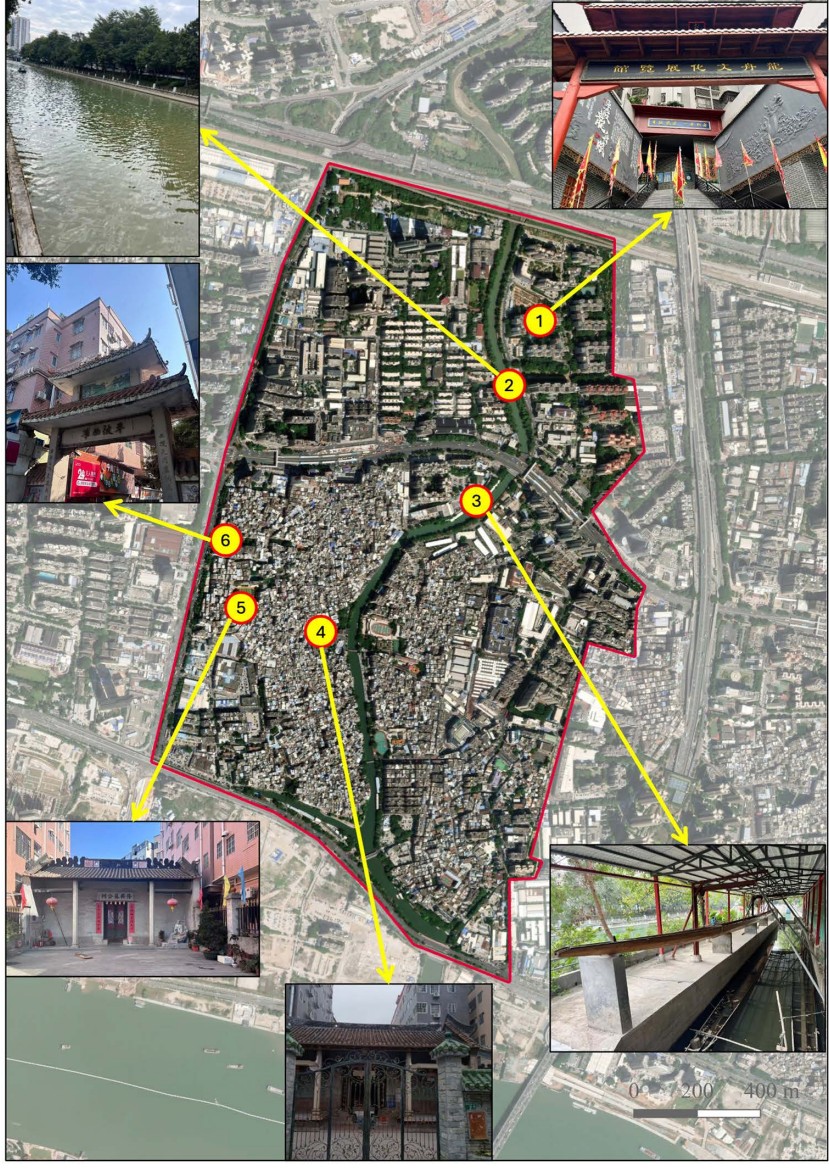

**Fig 3. Key Cultural Heritage Sites in Chebei Village (All photos were taken by the author, Xixi Tang.** This map is drawn with QGIS software and the map data is sourced from Natural Earth (https://www.naturalearthdata.com/)).

appropriate access points for visitor engagement, demonstrating how heritage tourism can be developed without displacing community ownership or cultural authenticity. The preservation and adaptive reuse of these facilities provides essential material foundations for the cultural experiences that constitute our identified topological invariant, while enabling the spatial reproduction processes that transform cultural assets into sustainable community resources.

The comprehensive nature of Chebei's cultural infrastructure, spanning practical facilities, ceremonial spaces, and interpretive venues, provides the material foundation for analyzing how ICH spaces maintain essential properties through radical environmental transformation while adapting to contemporary tourism development opportunities.

## Data collection methods and systematic approach

Data collection occurred systematically over two complete festival cycles (May-July 2022 and May-June 2023), employing multiple qualitative methods to ensure comprehensive data triangulation and rich empirical insights. The extended temporal scope enabled observation of seasonal variations, festival preparation cycles, and year-round cultural maintenance activities.

**Phase 1: Preliminary investigation and stakeholder mapping (May-July 2022).** Initial fieldwork focused on comprehensive stakeholder identification, preliminary observation of festival activities, and establishment of research relationships. This phase involved 20 exploratory interviews, extensive festival observation, and comprehensive archival research (Supporting Information S1 File Comprehensive Interview Guide). Key activities included mapping stakeholder networks, documenting festival procedures, and identifying theoretical sampling criteria for intensive fieldwork.

**Phase 2: Intensive fieldwork and theoretical development (May-June 2023).** The second phase involved 18 additional targeted interviews selected through theoretical sampling, extended participant observation across diverse settings, and systematic document analysis. Interview selection prioritized maximum variation sampling to capture diverse perspectives across stakeholder groups while ensuring theoretical saturation of emerging categories.

**In-depth Interviews (N = 38):** Semi-structured interviews with diverse stakeholders were conducted using theoretical sampling principles. Initial interviews with key stakeholders revealed the importance of including marginal voices, leading to expanded sampling that included migrants, tourists, and cultural entrepreneurs. Interviews ranged from 45–120 minutes, conducted in Mandarin or Cantonese based on participant preference, with simultaneous translation when necessary.

Interview protocols evolved through multiple iterations, beginning with broad questions about festival experience and gradually focusing on specific theoretical concerns emerging from data analysis. Key themes included personal history with dragon boat tradition, perceptions of continuity and change, roles in festival organization, impacts of urbanization and tourism, and visions for cultural futures. Photo elicitation using historical and contemporary festival images stimulated discussion about transformation processes while enabling participants to articulate complex temporal relationships.

Participant distribution included village leaders and community staff (N = 7), ancestral hall administrators and dragon boat association members (N = 12), ICH inheritors and cultural practitioners (N = 5), local residents (N = 4), migrant tenants (N = 3), local entrepreneurs (N = 3), and tourists/cultural enthusiasts (N = 4). This distribution ensured representation across stakeholder groups while maintaining focus on cultural practitioners and community members most directly involved in heritage transmission. These participants were broadly grouped into: (1) Traditional local stakeholders (Codes 1a, 1b, 1c, 1d, 1e), totaling 21 individuals (55.3%), who are core actors in organization, transmission, and interpretation. (2) External and associated populations (Codes 1f, 1g, 1h, 1i, 1j), totaling 17 individuals (44.7%), representing new immigrants, economic participants, and external observers crucial for observing the negotiated dynamics of the ICH space.

**Participant Observation (120 + hours):** Extensive ethnographic observation recorded spatial practices, social interactions, and meaning-making processes across diverse settings. Observation sites included dragon boat training sessions, festival preparation activities, ritual performances, tourist interactions, museum operations, and community meetings. Detailed field notes combined descriptive observations with analytical reflections, enabling identification of tacit knowledge and embodied practices difficult to capture through interviews alone.

Specific observation activities included documenting elderly craftsmen teaching boat repair techniques to younger generations, recording dragon head dotting ceremonies and associated ritual protocols, participating in communal meals that provided insights into cultural transmission processes, and observing tourist-resident interactions during festival periods. Photography and video recording, conducted with appropriate permissions, captured visual dimensions of spatial transformation and cultural performance.

Participant observation, such as engaging in dragon boat training, "rising dragon" rituals, and communal "dragon boat rice," was not only for collecting objective facts but also for immersing the researchers in the cultural context. This holistic, situated understanding and key scene impressions provided a "sensory basis" and "contextual clues" during the open

coding phase, helping to capture tacit cultural logic and emotional values. In the axial and selective coding phases, it facilitated the understanding of the internal relationships between categories.

**Document Analysis:** Comprehensive documentary materials provided historical depth and contextual understanding essential for temporal analysis. Government documents included urban planning reports, heritage policies, and tourism development strategies, revealing official approaches to village development and cultural preservation. Historical materials encompassed local gazetteers, genealogical records, and archival photographs documenting long-term transformation processes. Contemporary media analysis included news reports, social media content, and promotional materials displaying current representations and ongoing debates about heritage development.

Museum exhibitions provided particularly rich analytical material, materializing community memory through artifact selection, narrative construction, and spatial arrangement. Analysis of exhibition design revealed how communities conceptualize their own transformation, highlighting perceived continuities within apparent change and identifying community priorities for heritage preservation and presentation.

## Data analysis procedures and systematic coding

Analysis followed grounded theory's three-stage coding process, supported by qualitative data analysis software (NVivo 12) to ensure systematic and transparent analytical procedures (Table 1).

**Open Coding Phase:** Initial line-by-line analysis systematically fragmented data into discrete conceptual units (Supporting Information S2 File Detailed Grounded Theory Coding Structure). Researchers identified 127 initial codes capturing diverse phenomena including "dragon boat complex" (deep emotional attachment to tradition), "creek pollution memory" (collective environmental degradation memories), "tourist photography interference" (visitor behavior affecting cultural practices), "youth skill transmission" (intergenerational knowledge transfer), "clan economic strength" (traditional resource mobilization), and "ritual compression" (ceremonial adaptation to contemporary constraints).

Through constant comparative analysis, relationships among codes were systematically identified, forming preliminary conceptual categories. This process revealed patterns of cultural persistence, spatial transformation, and social adaptation that informed subsequent theoretical sampling and focused data collection.

**Axial Coding Phase:** Researchers reorganized data around emerging categories, exploring relationships through paradigm models examining causal conditions, contextual factors, action/interaction strategies, intervening conditions, and

**Table 1. Example of correspondence between core interview questions and coding categories.**

| Core Interview Question Area | Example of Corresponding Main/Subcategories | Explanation |
|---|---|---|
| Evolution and History of Dragon Boat Scenery | ICH Evolution (Main Category)<br>Ritual, Folk Activity, Cultural Symbol (Subcategories) | Questions focus on change, corresponding to the "Evolution" dimension |
| Organization and Execution of Dragon Boat Scenery | Power Space (Main Category)<br>Internal Collective Space, Multi-party Co-construction Space (Subcategories) | Questions involve organizational actors and rules, corresponding to the "Power" dimension |
| Protection and Perception of Dragon Boat Culture | ICH Experience (Main Category)<br>Sense of Ritual, Entertainment (Subcategories)<br>Cultural Space (Main Category)<br>Spatial Memory (Subcategory) | Questions focus on value and feeling, corresponding to "Experience" and "Cultural Memory" |
| Economic Environment and Impact of Dragon Boat Scenery | Capital Space (Main Category)<br>Spatial Value-add, Spatial Practice (Subcategories)<br>ICH Presentation (Main Category)<br>Economic Environment (Subcategory) | Questions involve costs, benefits, and economic activity, corresponding to "Capital" and "Economic Environment" |
| Dissemination and Innovation of Dragon Boat Culture | Cultural Space (Main Category)<br>Spatial Integration, Spatial Reconstruction (Subcategories) | Questions focus on modern adaptation and dissemination, corresponding to cultural "Integration" and "Reconstruction" |

consequences. This systematic analysis revealed how spatial transformation operates through interrelated cultural, political, and economic processes operating at multiple scales simultaneously.

Six major categories emerged through this process: ICH Evolution (documenting cultural development through temporal layers), ICH Presentation (analyzing material conditions for cultural practice), ICH Experience (identifying invariant experiential qualities), Cultural Space Reproduction (tracking narrative and spatial innovations), Power Space Reproduction (examining authority reconfigurations), and Capital Space Reproduction (analyzing economic value creation). Each category encompassed three subcategories, creating the 18-component analytical structure.

**Selective Coding Phase:** Integration around the core category "cultural experience as topological invariant" provided theoretical coherence and analytical focus. This central finding emerged through systematic analysis of elements that remained constant despite dramatic environmental changes—not physical forms or specific practices but experiential qualities connecting participants across generations and cultural backgrounds.

Selective coding refined relationships among categories, revealing the three-dimensional spatial production model and establishing causal relationships between tourism development, spatial reproduction mechanisms, and cultural continuity. The systematic nature of this analysis enabled clear differentiation between correlation and causation in observed relationships, strengthening theoretical validity.

**Theoretical Saturation and Validation:** Theoretical saturation was achieved when new data consistently confirmed rather than modified existing theoretical categories, with final interviews providing validation rather than novel insights. Member checking with key informants validated preliminary findings and incorporated community feedback into analytical refinements. Regular peer debriefing sessions challenged analytical assumptions and ensured theoretical rigor.

### Research quality and ethical considerations

Research quality was ensured through multiple systematic strategies designed to enhance credibility, transferability, dependability, and confirmability. Methodological triangulation using multiple data sources (interviews, observations, documents) and investigator triangulation involving multiple researchers enhanced analytical credibility. Thick description of research context and systematic documentation of analytical procedures support transferability assessments and replication efforts.

Audit trails comprehensively documented analytical decisions, enabling retrospective review of theory development processes and ensuring analytical transparency. External auditing through academic peer review provided independent assessment of analytical rigor and theoretical coherence. These quality assurance measures ensure that findings represent valid theoretical contributions rather than researcher bias or methodological artifacts.

This study was reviewed and approved by the Academic Committee of Guangdong University of Education (Ethical Review Exemption Certificate, Ref: 20251028). The committee determined that the research, which involved interviews and participant observation with non-vulnerable adult populations, constituted 'minimal risk research' and was exempt from routine supervision. All participants were adults. Prior to interviews, the research purpose, methodology, data usage, privacy protection measures, and participant rights (including unconditional withdrawal) were fully disclosed. Verbal consent was obtained from each participant, as this method was reviewed and approved by the ethics committee for this low-risk study. All interview data have been anonymized to protect participant privacy and ensure confidentiality.

Participant confidentiality was ensured through systematic anonymization using alphanumeric coding systems rather than personal identifiers. Primary interview data were coded as 1a-1j, with 1a representing community street office staff (where 1a-1 indicates the first community street office staff interview, 1a-2 the second, and so forth), 1b representing ancestral hall dragon boat association staff, 1c representing dragon boat cultural exhibition hall staff, 1d representing local residents, 1e representing ICH inheritors, 1f representing migrant tenants, 1g representing business owners, 1h representing vendor stall operators, 1i representing tourists, and 1j representing dragon boat culture enthusiasts.

Secondary materials received coding as 2a-2e, representing government official documents (2a), video-converted textual materials (2b), website news content (2c), dragon boat cultural exhibition hall display materials (2d), and related published sources (2e). This systematic coding ensures traceability while protecting participant identities and enabling systematic analysis across data sources.

Community benefit sharing included providing research reports to village leadership, supporting museum development initiatives, and facilitating cultural exchange projects. These activities ensured that research generated tangible benefits for the studied community while maintaining academic independence and analytical rigor.

## Results

### The "2-6-18" generative logic of ICH space structure

Our systematic grounded theory analysis revealed a comprehensive "2-6-18" generative logic underlying Chebei's ICH space dynamics (Table 2). This refers to 2 major analytical dimensions (topological structure and spatial reproduction), 6 main categories (ICH Evolution, ICH Presentation, ICH Experience, Cultural Space Reproduction, Power Space Reproduction, Capital Space Reproduction), and 18 subcategories that systematically capture the complexity of urban village ICH spaces while providing analytical coherence for cross-case application.

This generative logic operates simultaneously across multiple scales, from individual experiential responses to regional cultural networks, demonstrating how micro-level cultural practices connect to macro-level urban transformation processes. The systematic nature of this structure enables both detailed analysis of specific cultural phenomena and broader theoretical application to diverse heritage tourism contexts.

### The topological structure of dragon boat heritage space

Our analysis demonstrates how Chebei's dragon boat heritage space exhibits topological properties through three interconnected dimensions that operate simultaneously across multiple temporal and spatial scales. These findings reveal

**Table 2. The "2-6-18" Generative logic of ICH space structure and reproduction.**

| Major Section | Main Category | Subcategory | Description | Analytical Scale |
|---|---|---|---|---|
| Topological Structure | ICH Evolution | Historical Imprints | Transformation from ritual to contemporary symbol | Temporal/Regional |
| | | Ritual Refinement | Continuous adaptation of traditional practices | Community/Individual |
| | | Symbolic Density | Multiple meanings compressed in cultural forms | Cultural/Cognitive |
| | ICH Presentation | Physical Environment | Material conditions enabling practice | Spatial/Local |
| | | Economic Environment | Resource base supporting activities | Economic/Community |
| | | Social Environment | Networks and relationships | Social/Regional |
| | ICH Experience | Embodied Knowledge | Skills transmitted through practice | Individual/Somatic |
| | | Collective Effervescence | Shared emotional states | Social/Temporal |
| | | Temporal Connection | Links across generations | Temporal/Cultural |
| Spatial Reproduction | Cultural Space | Narrative Innovation | New forms of cultural storytelling | Discursive/Media |
| | | Spatial Reinterpretation | Physical spaces gaining new meanings | Spatial/Symbolic |
| | | Experience Design | Creating participatory opportunities | Individual/Commercial |
| | Power Space | Hybrid Governance | Traditional-modern authority combinations | Institutional/Legal |
| | | Inclusive Participation | Expanding beyond clan boundaries | Social/Political |
| | | Multi-scalar Networks | Local-global connections | Scalar/Organizational |
| | Capital Space | Asset Transformation | Converting resources for cultural use | Economic/Material |
| | | Value Chain Development | Economic linkages around heritage | Economic/Network |
| | | Symbolic Capital Conversion | Prestige becoming economic value | Cultural/Economic |

how ICH spaces maintain essential characteristics through radical urban transformation while adapting to contemporary contexts.

**ICH evolution: The formation of symbolic density through temporal folding.** The dragon boat tradition's evolution from sacred ritual to multilayered cultural symbol illustrates topological folding processes operating across historical time scales. Rather than linear progression where new forms replace earlier elements, each historical phase adds interpretive layers creating increasingly dense symbolic space capable of supporting multiple simultaneous meanings.

## Cosmological foundation and sacred persistence

Archaeological evidence and elder testimonies reveal dragon boat racing's origins in agricultural rituals seeking divine protection from water deities. This cosmological foundation persists within contemporary practice despite surface secularization. As one ritual specialist explained: "People today see celebration, but originally this was survival. The dragon king controlled rain and floods. Racing boats showed our strength, pleased the gods, brought good harvest. My grandfather fasted three days before touching the dragon head—it was that sacred" (1e-1).

This cosmological layer remains present through what we term "ritual compression"—the condensation of elaborate ceremonial sequences into symbolic gestures that maintain essential spiritual meanings within contemporary time constraints. Pre-race blessing ceremonies continue, boats undergo "eye-opening" rituals believed to animate dragon spirits, and races maintain alignment with agricultural cycles despite participants' urban occupations. The sacred dimension persists not through unchanged replication but through adaptational continuity that preserves relational meanings within altered forms.

## Social consolidation and network persistence

Ming-Qing period historical documents demonstrate dragon boat events evolving into comprehensive regional social systems extending far beyond individual village boundaries. Chebei's designation as "official scenic spot" reflected its emergence as a cultural node linking villages through complex webs of reciprocal obligations based on kinship, marriage alliances, and historical relationships. A clan association leader described the persistence of these social networks: "Each village had its own boats, but we were all connected. You must visit certain villages based on ancestry, marriage, or historical alliance. It's like a map of relationships written on water. Even today, we follow these routes" (1b-2).

This social architecture persists remarkably intact despite urban transformation that has severed many physical village connections. Contemporary GPS tracking of boat routes reveals how current races continue to trace historical relationship patterns, creating what participants term "water roads" (shuilu) that link past and present geographies. These persistent social networks demonstrate topological continuity where relational structures endure through radical environmental transformation, maintaining cultural integrity across spatial and temporal scales.

## Cultural symbolization and interpretive flexibility

Contemporary dragon boat culture operates as a complex semiotic system encoding multiple layers of meaning that enable broad cultural participation while preserving deep traditional significance. Museum displays demonstrate how boat decorations communicate village identity markers, racing formations express social hierarchies, and ritual protocols embody moral values transmitted across generations. A cultural educator articulated this symbolic complexity: "Every element tells stories. The dragon's expression shows village character—fierce for warrior clans, benevolent for scholarly families. Colors represent the five elements. Drum rhythms encode rowing instructions. It's a living library" (1c-1).

This symbolic density enables cultural continuity through interpretive flexibility that accommodates diverse participant backgrounds and knowledge levels. Younger participants may not fully comprehend cosmological significance but readily understand identity markers and social belonging dimensions. Tourists perceive spectacular performance while practitioners experience layered spiritual and social meanings. The symbol system's redundancy—multiple elements encoding

similar messages—ensures cultural transmission despite partial understanding, demonstrating topological robustness through meaning preservation across interpretive diversity.

**ICH presentation: Material conditions enabling cultural practice.** Analysis reveals three environmental dimensions—physical, economic, and social—that create material conditions enabling dragon boat culture's persistence within urban transformation. These dimensions operate interdependently, with changes in one affecting possibilities within others, demonstrating systemic relationships essential for cultural continuity.

## Physical environment and ecological restoration

Chebei Creek's material qualities fundamentally shape cultural possibilities through providing essential infrastructure for large-scale dragon boat activities. The waterway's 800-meter straightaway with 50-meter width provides increasingly rare urban space capable of accommodating competitive races involving hundreds of boats and tens of thousands of spectators. However, industrial development severely threatened this physical foundation during the 1990s economic expansion period.

Environmental degradation reached crisis levels that directly threatened cultural survival: "In the 1990s, the creek was black sewage. Factories dumped chemicals, residents threw garbage. No dragon boat could enter—it would corrode! We thought our tradition would die with the water" (1d-3). This crisis demonstrates the material dependencies of cultural practice—tradition cannot survive solely through memory or symbolic preservation without adequate physical spaces for enactment and embodied learning.

Comprehensive environmental restoration costing over 100 million RMB reversed ecological degradation through sewage interception, sediment dredging, and systematic rehabilitation programs. Water quality improvement from Class V (severely polluted) to Class III (suitable for human contact) literally enabled cultural revival, demonstrating causal relationships between environmental conditions and cultural possibility. Contemporary infrastructure enhancements support cultural activities while respecting traditional spatial requirements, showing how modernization can strengthen rather than undermine tradition when guided by cultural understanding.

## Economic environment and resource mobilization

Dragon boat culture requires substantial financial resources that create ongoing sustainability challenges within contemporary urban contexts. Boats cost 100,000–200,000 RMB each, annual maintenance exceeds 50,000 RMB per vessel, and festival expenses reach 500,000 RMB per association annually. These costs necessitate robust economic foundations that Chebei's unique collective economy provides: "Our ancestral halls own properties—shops, apartments, warehouses. Rental income funds dragon boat activities. Without collective economy, only wealthy villages could maintain traditions" (1b-1).

This economic model demonstrates path-dependent institutional development where historical property arrangements enable contemporary cultural practice. Villages lacking collective assets face significant challenges maintaining traditions despite strong cultural commitment, illustrating causal relationships between economic foundations and cultural sustainability. However, Chebei's model confronts pressures as land values escalate and urban redevelopment threatens traditional income sources, necessitating adaptive strategies including tourism development to generate alternative revenue streams while maintaining cultural priorities.

## Social environment and demographic transformation

Dragon boat culture's social reach extends far beyond traditional village boundaries, encompassing remarkable demographic diversity that challenges conventional assumptions about cultural authenticity and community membership. Participant observation revealed active participation from multi-generational local families, long-term migrants, young professionals, and international tourists temporarily united through shared cultural experience.

Migration patterns fundamentally reshape social dynamics within urban villages. Chebei's 80% migrant population might theoretically dilute traditional culture, yet many migrants actively embrace dragon boat activities as sources of community belonging: "I'm from Hunan, lived here fifteen years. My son trains with the youth team. During festival, we're all Chebei people. This gives belonging you don't find in modern apartments" (1f-2).

This inclusive dynamic contrasts sharply with exclusionary practices observed in some traditional cultural contexts. Dragon boat culture's emphasis on collective effort and embodied skill development over individual ancestry or cultural knowledge enables incorporation of newcomers through participatory learning. Youth teams particularly demonstrate cultural fusion processes where local and migrant children train together, learn Cantonese through dragon boat chants, and form friendships that transcend geographic origins, creating new forms of community identity that maintain cultural continuity while embracing demographic change.

**ICH experience: Identifying the topological invariant.** Through systematic analysis of elements that persist despite radical environmental transformation, we identified "cultural experience" as the core topological invariant maintaining cultural continuity across time, space, and social change. This experiential quality comprises three interrelated elements that enable cultural transmission and community formation independent of specific material forms or social structures.

## Embodied knowledge and somatic transmission

Dragon boat culture transmits primarily through bodily practice rather than verbal instruction, creating forms of cultural knowledge that prove remarkably resistant to disruption through environmental change. Training observations revealed how paddling techniques, rhythm coordination, and spatial awareness develop through repetitive collective action that inscribes cultural patterns within participants' muscle memory and proprioceptive awareness. A veteran coach articulated this embodied transmission process: "You cannot learn from books. The water teaches you—how current pulls, wind pushes, waves rock. Your body remembers grandfather's movements even if mind forgets stories" (1e-2).

This embodied transmission demonstrates exceptional robustness across cultural and linguistic boundaries. Youth who speak minimal Cantonese and possess limited traditional cultural knowledge still acquire deep competence through sustained practice, suggesting that somatic learning bridges cultural gaps that verbal transmission cannot cross. Even tourist participants report powerful experiential connections despite brief exposure: "I joined a trial session as cultural tourism. But when you're paddling in sync, drum pounding, everyone shouting—something primal awakens. I understand why they preserve this" (1i-3).

## Collective effervescence and shared emotional states

Following Durkheim's concept of collective effervescence, dragon boat events generate intense emotional states through synchronized ritual action that transcends individual cultural backgrounds and creates temporary communities united by shared experience [28]. Dragon boat events produce what participants describe as "dragon energy" (longqi)—heightened collective awareness emerging from coordinated movement, rhythmic drumming, and synchronized vocalization. Field observation documented this phenomenon: "Crowd energy becomes palpable as boats align at starting positions. Drummers begin synchronized beating, 40 boats with 40 drums creating thunderous rhythm. Paddlers shout in unison, bodies moving as one organism. Spectators spontaneously cheer. Air feels electric with anticipation. Individual boundaries dissolve into collective force."

This experiential intensity transcends cultural boundaries and linguistic differences. Migrants, tourists, and local residents report similar sensations despite different interpretive frameworks for understanding their experience. The physiological and emotional response itself—rather than specific cultural meanings attached to the experience—serves as a cultural constant enabling community formation and cultural transmission across diverse populations, demonstrating how embodied experience can maintain cultural continuity through demographic transformation.

## Temporal connection and transgenerational bonding

Participants consistently describe profound temporal experiences during dragon boat activities that collapse conventional linear time sequences, creating what we term "thick time" where historical layers become simultaneously present rather than sequentially ordered. This temporal experience manifests through various mechanisms including handling implements used by ancestors, performing movements unchanged across generations, and visiting sites of historical significance. A young participant articulated this temporal collapse: "When I hold the paddle my great-grandfather carved, I feel him with me. Not ghost—more like time disappears. Past and present exist together on the water" (1d-8).

This temporal dimension distinguishes heritage experience from ordinary recreational activities and provides powerful motivation for cultural participation and transmission. The temporal experience persists whether participants use traditional wooden boats or modern fiberglass versions, suggesting that the invariant quality resides in relational and experiential dimensions rather than material authenticity. This finding challenges common heritage management assumptions about material preservation requirements while supporting emphasis on experiential continuity and participatory engagement.

Therefore, through the interpretation of ICH evolution, presentation, and experience, "cultural experience" is identified as the "topological invariant" in Chebei's Dragon Boat Scenery space. Its stability is manifested in the shared core experiential kernel. For example, while local residents emphasize intergenerational inheritance (1d-9) and traditional rituals, migrant tenants transcend kinship and geography to feel a broader identity ("This is our Chinese festival") (1d-6), and tourists resonate strongly with the spectacular scenes and cultural charm (1i-1, 1i-2). This spectrum precisely demonstrates that "cultural experience," as a stable kernel, possesses the topological connection point function to connect and cohere diverse social groups.

## Mechanisms of spatial reproduction across scales

Analysis of how dragon boat heritage space reproduces itself through urban transformation reveals three intersecting processes operating simultaneously through cultural, power, and capital dimensions at multiple scales from individual experience to regional networks (Table 3). These mechanisms demonstrate how tourism development can enable adaptive preservation when properly integrated with community dynamics and cultural priorities.

### Cultural space reproduction: From collective memory to heritage brand

Contemporary cultural reproduction involves active transformation of traditional collective memory into heritage brands suitable for tourism consumption while maintaining experiential authenticity and community ownership. This process

**Table 3. Research participant demographics and coding structure.**

| Code | Stakeholder Type | Number | Gender | Age Range | Years in Chebei | Key Insights |
|------|------------------|--------|--------|-----------|-----------------|--------------|
| 1a | Community Street Office Staff | 4 | 3M, 1F | 35-55 | 5-20 | Governance perspectives, policy implementation |
| 1b | Dragon Boat Association Members | 3 | 3M | 45-65 | Lifetime | Traditional authority, cultural transmission |
| 1c | Museum Staff | 2 | 1M, 1F | 30-45 | 3-10 | Heritage interpretation, visitor engagement |
| 1d | Local Residents | 12 | 7M, 5F | 25-75 | Lifetime | Community transformation, cultural continuity |
| 1e | ICH Inheritors | 2 | 2M | 55-70 | Lifetime | Ritual knowledge, traditional practices |
| 1f | Migrant Tenants | 6 | 4M, 2F | 25-45 | 2-20 | Cultural adaptation, community integration |
| 1g | Business Owners | 3 | 2M, 1F | 35-55 | 5-30 | Economic impacts, commercial adaptation |
| 1h | Vendor Stall Operators | 2 | 1M, 1F | 40-50 | 10-15 | Economic participation, festival commerce |
| 1i | Tourists | 3 | 2M, 1F | 25-40 | First visit | External perspectives, cultural experience |
| 1j | Cultural Enthusiasts | 1 | 1M | 35 | 15 | Cultural preservation, knowledge documentation |

illustrates topological unfolding—making implicit cultural knowledge explicit and accessible to broader audiences without fundamentally altering essential experiential properties or community relationships.

### Narrative innovation and media adaptation

Traditional dragon boat stories, previously transmitted orally within close community networks, now circulate through multiple media platforms reaching diverse audiences while maintaining cultural coherence and emotional resonance. The viral video "One Water, One Boat" exemplifies successful cultural translation that preserves experiential core while adapting surface forms for contemporary consumption: "We struggled explaining dragon boat culture to outsiders—too complex, too local. The video captures feeling without lecturing. Ten million views brought young people asking to join. Tradition needs new languages" (1a-3).

Systematic analysis reveals how successful narratives maintain experiential authenticity while employing contemporary aesthetic forms. The video integrates modern production techniques including drone footage, electronic music composition, and rapid editing sequences while structuring narrative content around traditional themes of water, community, and temporal continuity. Social media comment analysis demonstrates viewers responding primarily to emotional rather than informational content, suggesting that cultural experience transcends specific cultural knowledge or linguistic comprehension, supporting our findings about experiential invariance.

### Spatial reinterpretation and heritage infrastructure

Physical spaces undergo systematic reinterpretation that makes cultural significance visible and accessible to outsiders while preserving community functions and traditional spatial relationships. The dragon boat museum exemplifies this unfolding process through carefully designed exhibition spaces that translate tacit cultural knowledge into explicit interpretive frameworks: "Villagers donated 5,000 objects—photos, trophies, paddles, documents. Each tells family stories. We create exhibitions helping visitors understand what they see isn't just sport but life itself" (1c-2).

Museum design employs contemporary interpretive techniques including interactive displays, multimedia presentations, and immersive environmental design while respecting traditional spatial concepts and community priorities. The central hall deliberately reproduces ancestral hall architectural elements, positioning dragon boat artifacts as sacred community symbols rather than historical curiosities. Exhibition narrative flow follows traditional festival temporal sequences, enabling visitors to experience cultural rhythms and seasonal cycles. This spatial translation makes previously tacit community knowledge accessible to outsiders without reducing cultural complexity to simplified tourist narratives.

### Experience design and participatory engagement

Beyond passive heritage display, new cultural venues enable active participatory engagement that transmits cultural experience through embodied learning rather than purely cognitive interpretation. Dragon boat simulators enable visitors to experience paddling coordination and rhythm requirements. Traditional craft workshops teach decorative techniques and construction methods. Festival preparation activities invite carefully managed tourist participation in community cultural work. These innovations recognize experiential learning as essential for authentic cultural understanding: "Watching races provides spectacle. But culture lives in doing—feeling paddle weight, finding rhythm, sharing exhaustion. We design experiences transmitting feeling not just information" (1g-2).

### Power space reproduction: Negotiating authority across scales

Power relationships undergo complex reconfiguration as traditional authority structures adapt to contemporary governance requirements while maintaining cultural legitimacy and community control. This process reveals how power spaces can transform topologically—changing organizational configuration while preserving essential authority relationships and decision-making processes that protect community interests and cultural priorities.

 

## Hybrid governance and institutional innovation

Dragon boat associations exemplify governance innovation that successfully combines traditional cultural authority with modern legal requirements, creating hybrid institutions capable of operating within contemporary urban administrative systems while preserving community autonomy and cultural decision-making processes. Formally registered as civil society organizations under government oversight, these associations maintain internal organization through customary principles and traditional leadership selection: "Government requires democratic elections, financial transparency, safety standards. But we elect leaders from traditional dragon boat families who understand cultural requirements. Modern structure, traditional spirit" (1b-2).

This hybrid governance model enables legal operation within contemporary urban administrative frameworks while preserving cultural authority and community control over heritage decisions. Association leaders function as cultural brokers, mediating between government regulators who prioritize safety, legal compliance, and administrative efficiency, and community members who value traditional autonomy, cultural authenticity, and customary practices. Success requires continuous negotiation and adaptation—developing procedural compliance with bureaucratic requirements while maintaining community trust and cultural legitimacy through transparent decision-making and inclusive participation.

## Inclusive participation and community boundary evolution

Traditional clan-based organizational structures evolve toward inclusive membership models that reflect contemporary demographic realities while maintaining cultural priorities and community cohesion. Constitutional reforms enable non-surname community members to achieve full membership through cultural contribution and sustained participation, creating new categories including "cultural contributors" and "honorary paddlers" that recognize commitment over ancestry. A migrant member explained this transformational inclusion: "My family name isn't Chen, but I've paddled ten years, my children grew up here. The association voted me full member. Tradition expands embracing those who embrace it" (1f-4).

This inclusivity transformation maintains fundamental meritocratic principles where membership derives from demonstrated contribution and cultural commitment rather than inherited birthright, while adapting organizational boundaries to accommodate multicultural demographic reality. Tensions persist as some traditionalist community members resist perceived dilution of ancestral authenticity, yet pragmatist leaders recognize that cultural survival requires community expansion and demographic adaptation. The preserved invariant proves to be collective commitment to cultural practice and community welfare rather than exclusive bloodline continuity, demonstrating topological preservation of essential relationships through transformed membership criteria.

## Multi-scalar networks and global-local connections

Local dragon boat associations increasingly participate in regional, national, and international dragon boat organizations that provide resources, technical standards, competitive opportunities, and cultural recognition while strengthening rather than undermining local cultural identity and community autonomy. These network connections illustrate topological principles of maintaining local cultural properties while establishing beneficial distant relationships: "Joining the International Dragon Boat Federation brought technical standards, referee training, equipment upgrades. But we maintain Chebei characteristics—our rituals, our routes, our relationships. Global connection strengthens local identity" (1a-4).

Network participation demonstrates successful multi-scalar cultural strategy that contrasts with both isolated cultural localism and homogenizing cultural globalization. Chebei gains access to resources, competitive opportunities, and international recognition through global organizational integration while using these platforms to assert and celebrate distinctive local cultural identity. This approach enables cultural communities to benefit from global connections while maintaining local control over cultural priorities and community development directions.

### Capital space reproduction: Value creation and equitable distribution

Economic transformation from traditional subsistence agriculture through industrial production to emerging cultural economy demonstrates how capital spaces can undergo radical functional reconfiguration while supporting rather than undermining cultural continuity and community welfare. The developing heritage tourism economy illustrates both possibilities and potential tensions within cultural commodification processes, requiring careful management to ensure community benefit and cultural integrity.

### Asset transformation and cultural investment

Village collective assets undergo systematic functional transformation from agricultural to cultural production purposes, demonstrating adaptive resource utilization that honors ancestral legacy while meeting contemporary community needs. Former agricultural land now hosts dragon boat training facilities and festival infrastructure. Ancestral halls serve dual functions as community cultural centers and tourist interpretation venues. Waterfront properties support tourism-related businesses while maintaining community gathering spaces and cultural activities: "Our ancestors left land for rice growing. We honor them using land for culture growing. Dragon boat tourism brings more income than factories, preserves what factories would destroy" (1a-2).

This transformation requires careful balance between commercial development and cultural preservation priorities. Excessive commercialization threatens cultural authenticity and community ownership, while insufficient economic development cannot compete with alternative land uses or generate adequate resources for cultural maintenance. Successful examples integrate commercial and cultural functions through dragon boat-themed restaurants housed in traditional buildings, craft workshops operating in converted warehouses, and viewing platforms that serve both tourist accommodation and community gathering purposes during festivals and daily cultural activities.

### Value chain development and economic integration

Heritage tourism creates complex value chains that link cultural production to economic benefits across multiple scales, generating direct revenue for cultural maintenance while creating broader economic opportunities for community members and local businesses. Primary cultural attractions including races, museum visits, and craft demonstrations generate secondary economic activity through accommodation, dining, shopping, and transportation services that create beneficial multiplier effects: "Festival week, my restaurant serves 1,000 extra customers daily. I hire temporary staff, buy more supplies, everyone benefits. Culture drives economy, economy supports culture" (1g-1).

However, value distribution remains contested and requires ongoing management to ensure equitable community benefit. External tour operators often capture significant economic value while contributing minimally to cultural maintenance or community welfare. Street vendors may compete unfairly with established local businesses. Increased tourism generates costs including traffic congestion, parking shortages, and crowding that impose burdens on residents who may not directly share tourism benefits. Addressing these distributional challenges requires governance mechanisms that ensure tourism development generates equitable community benefits while protecting residents from negative externalities.

### Symbolic capital conversion and prestige economics

Provincial ICH designation brings cultural prestige that converts systematically to economic value through various mechanisms including government investment attraction, cultural grant qualification, and enhanced marketing credibility. Official heritage recognition provides powerful marketing advantages: "The ICH certificate hangs in every dragon boat restaurant. Tourists trust officially recognized culture. One piece of paper worth millions in marketing value" (1g-3).

This symbolic capital operates subtly yet powerfully across multiple economic domains. Property values increase significantly near recognized cultural sites. Local businesses adopt dragon boat imagery and cultural themes in branding

strategies. Young professionals cite cultural vibrancy as important factors when selecting Chebei for residence. The village's cultural reputation creates economic premiums that extend far beyond direct tourism revenue, demonstrating how cultural investment generates broad-based economic returns that benefit the entire community while supporting ongoing cultural maintenance and development.

## COVID-19 as topological resilience test

The COVID-19 pandemic provided an unprecedented natural experiment testing cultural resilience under extreme disruption, forcing complete cessation of dragon boat activities during 2020's strict lockdown measures. This crisis revealed both vulnerabilities and adaptive capacities within the topological framework, demonstrating how cultural invariants can persist through radical deformation of practice modes while maintaining essential community relationships and cultural transmission processes.

Physical gatherings—seemingly fundamental to dragon boat culture—became impossible under pandemic restrictions, yet communities demonstrated remarkable topological flexibility in maintaining cultural connections and knowledge transmission. Virtual drumming sessions enabled rhythm training continuation across physical separation. Social media platforms facilitated technique video sharing and remote instruction. Online ceremonies maintained ancestral honoring traditions and seasonal observances. Young participants created "Dragon Boat Simulator" games that spread cultural knowledge while providing entertainment during isolation periods.

Most significantly, the forced hiatus intensified rather than diminished appreciation for embodied cultural practice. Post-lockdown participation surged beyond pre-pandemic levels, with increased youth engagement and stronger community commitment to cultural maintenance: "Missing dragon boat made us realize what we almost lost. Now every paddle stroke feels precious" (1d-12).

This pandemic response demonstrates topological resilience—the capacity to maintain essential cultural properties through radical deformation of surface practice modes. The cultural invariant of collective experience and embodied knowledge manifested through digital platforms and individual practice while community members awaited physical reunion, suggesting that cultural continuity depends more on relational commitment and experiential knowledge than specific material arrangements or gathering formats.

## Toward integrated spatial reproduction: Flagship ICH-Led cultural tourism development

Our analysis reveals an emerging comprehensive model we term "flagship ICH-led cultural tourism spatial reproduction"—an integrated approach that leverages distinctive cultural assets to drive comprehensive urban regeneration while maintaining community identity, cultural authenticity, and resident quality of life. This model demonstrates how tourism development can facilitate rather than undermine cultural continuity when properly structured around community priorities and cultural values.

## Spatial integration strategies and distributed development

Unlike conventional heritage tourism approaches that create isolated tourist attractions separate from community life, Chebei develops distributed cultural infrastructure woven throughout existing village fabric that enables authentic cultural experience while maintaining normal community functions. Dragon boat cultural elements appear throughout everyday spaces including paddle-shaped street signage, dragon-scale paving patterns, and boat-inspired architectural details that create cultural coherence without displacing residential or commercial activities. A planning official explained this philosophy: "We rejected theme park models—one big tourist zone separate from real life. Instead, culture everywhere, tourism everywhere, community everywhere. Visitors experience living culture not museum displays" (1a-1).

This distributed approach maintains community ownership and control while accommodating visitor access and cultural interpretation needs. Sacred spaces remain primarily community-focused while providing respectful viewing opportunities during appropriate occasions. Commercial development concentrates in designated areas that preserve residential tranquility and traditional community gathering spaces. The comprehensive strategy recognizes that sustainable heritage tourism requires careful balance between visitor accommodation and community integrity, ensuring that tourism enhances rather than displaces community life and cultural practice.

### Temporal extension and year-round programming

Moving beyond traditional festival-dependent tourism that concentrates economic activity within brief seasonal periods, Chebei develops comprehensive year-round programming that maintains cultural vitality and economic sustainability while providing diverse engagement opportunities for both community members and visitors. Regular activities include weekend dragon boat training exhibitions, monthly cultural workshops, seasonal ritual observations, and educational programs that connect cultural practice to contemporary life: "Depending on one festival means 360 dead days. We create reasons for visiting year-round—see training, learn crafts, taste cuisine, understand culture. Every visit builds toward festival climax" (1a-3).

This temporal extension strategy serves multiple interconnected functions across different scales. Economically, it smooths revenue flows and justifies infrastructure investment while providing stable employment for community members. Culturally, it maintains practitioner engagement and skill development throughout the year while enabling deeper visitor understanding of cultural complexity. Socially, it provides regular opportunities for community interaction and cultural transmission beyond intense festival periods. The approach recognizes culture as lived practice requiring continuous maintenance rather than periodic performance, ensuring authentic cultural experience while supporting sustainable economic development.

## Discussion

### Bridging topology and cultural heritage studies: Theoretical innovation

This study addresses fundamental theoretical limitations in heritage tourism scholarship by introducing topology as an analytical framework that successfully transcends traditional preservation-development dichotomies while providing robust tools for understanding cultural persistence within transformation processes. Our topological approach demonstrates that cultural change need not constitute rupture leading to authenticity loss, but rather can represent continuous deformation that potentially preserves essential relational properties and experiential qualities that define cultural identity and community belonging.

The theoretical bridging between mathematical topology and cultural analysis operates through several innovative mechanisms. First, topological invariance provides conceptual framework for understanding cultural continuity that transcends material fixity and enables recognition of cultural persistence through environmental transformation. Cultural practices maintain essential characteristics not through unchanging physical forms but through preserved relational structures, experiential qualities, and community relationships that adapt to contemporary conditions while maintaining core identity.

Second, the topological concepts of "folding" and "unfolding" offer powerful metaphorical tools for analyzing how cultural meaning becomes compressed within contemporary spaces during transformation processes (folding) and subsequently made explicit and accessible through tourism development and heritage interpretation (unfolding). This conceptual framework enables analysis of cultural dynamics as continuous transformation rather than binary preservation or loss, providing more nuanced understanding of heritage adaptation processes.

The identification of "cultural experience" as core topological invariant proves particularly significant for heritage management theory and practice. Rather than focusing on preserving supposedly immutable material forms or strictly replicating historical practices, this framework emphasizes sustaining deeper relational and embodied meanings that define

heritage participation and cultural transmission. This provides more flexible, realistic, and culturally sensitive approaches to authenticity assessment within dynamic cultural landscapes, acknowledging both living heritage fluidity and adaptation requirements for contemporary cultural survival.

However, it must be pointed out that this process is not a linear development devoid of tension. Based on critical perspectives from international ICH tourism research, this study identifies several potential risks. First, in the branding and symbolic production of "Cultural Space," there is a risk that the profound ritualistic connotations and clan emotions of the dragon boat culture will be simplified into consumable visual symbols and tourism labels, facing the challenge of "cultural flattening and hollowing out." Second, within the multi-party construction of "Power Space," although community participation is emphasized, the negotiation among government guidance, capital investment, and spontaneous community forces always exists. How the cultural identity and rights of "new villagers," such as migrant tenants, can be substantively embodied in the ICH space still requires more effective institutional guarantees to prevent the "weakening of community agency." Finally, driven by the value-added logic of "Capital Space," there is a need to be vigilant against commercial interests excessively eroding cultural authenticity, preventing the Dragon Boat Scenery from degenerating from a folk ritual that unites clan emotions into a pure tourism attractor and tool for economic growth. Therefore, future "flagship ICH-led cultural tourism spatial reproduction" must be built on a foundation of critical reflection and cultural sensitivity.

Current heritage tourism frameworks, whether based on Lefebvre's spatial production theory, Harvey's time-space compression analysis, or Smith's authorized heritage discourse critique, share common analytical limitation: difficulty conceptualizing simultaneous continuity and change within cultural systems [39]. Lefebvre's influential spatial triad, though revolutionary in revealing how space is actively produced through social processes rather than passively experienced, tends to position transformation against preservation in dialectical opposition. While his framework excels at explaining how capital and power produce new spatial configurations, it struggles to address how cultural meaning and community relationships persist through radical spatial reconstruction and demographic transformation.

### Empirical contributions and analytical framework development

Our empirically-grounded "culture-power-capital" three-dimensional framework extends spatial production theory to account for the specific complexities of ICH spaces within urban village contexts, demonstrating that heritage reproduction involves not solely cultural phenomena but profound interaction with economic realities and power distribution patterns operating at multiple scales simultaneously(**Fig 4**). This tripartite analytical framework reveals how tourism development acts as powerful catalyst reshaping all three dimensions while potentially maintaining topological properties through careful community-controlled management.

The systematic "2-6-18" generative logic provides comprehensive analytical structure for understanding ICH space dynamics while remaining sufficiently flexible for application across diverse cultural contexts and heritage tourism situations. This empirically-derived framework captures both micro-level cultural transmission processes and macro-level urban transformation dynamics, enabling analysis that connects individual experiential responses to regional development patterns and policy frameworks.

The causal relationships identified within this framework distinguish between correlation and causation in heritage tourism development. Environmental restoration directly enables cultural revival (causal relationship), while cultural tourism development correlates with but does not necessarily cause community social cohesion (correlational relationship requiring mediating factors). Tourism revenue generation can causally support cultural maintenance when properly managed through community-controlled mechanisms, while tourism development may correlate with cultural change without necessarily causing cultural loss or degradation.

These distinctions prove crucial for heritage tourism policy and management because they enable targeted interventions that strengthen causal mechanisms supporting cultural continuity while managing correlational relationships that may generate either positive or negative outcomes depending on implementation approaches and community involvement levels.

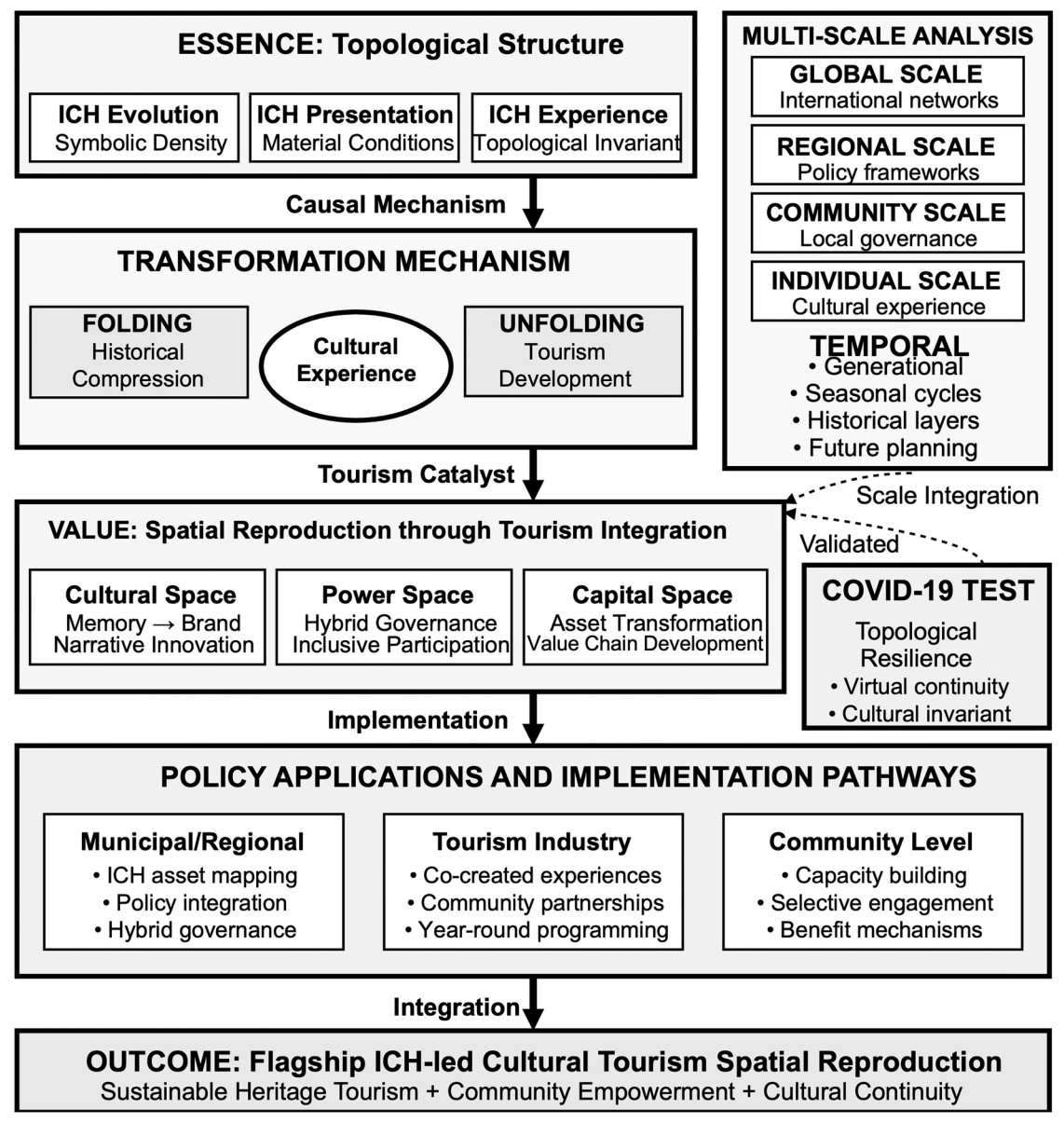

**Fig 4. ICH Space Transformation Logic.**

## Scale integration and multi-level analysis

This research addresses the critical need for heritage tourism analysis that operates effectively across multiple spatial and temporal scales, from individual embodied experience to regional cultural networks and from seasonal festival cycles to multi-generational cultural transmission processes. The topological framework enables scale integration by focusing on relational properties that persist across scalar transformations while adapting to scale-specific conditions and requirements.

Individual-level analysis reveals how embodied knowledge and somatic learning enable cultural transmission that transcends linguistic and cultural boundaries, supporting inclusive community development while maintaining cultural

authenticity and traditional knowledge systems. Community-level analysis demonstrates how hybrid governance structures enable adaptation to contemporary institutional requirements while preserving traditional authority relationships and cultural decision-making processes.

Regional-level analysis shows how local cultural communities can benefit from global network participation while maintaining distinctive cultural identity and community autonomy over development priorities. Temporal analysis reveals how cultural practices adapt across generational change, seasonal cycles, and historical transformation while preserving essential experiential qualities and community relationships that define cultural identity and belonging.

This multi-scalar approach enables heritage tourism strategies that operate effectively across different analytical levels while maintaining coherence and cultural integrity. Local community priorities guide development directions while regional and global connections provide resources and opportunities for cultural sustainability and community welfare enhancement.

## Policy implications and implementation strategies

Our findings generate specific, actionable policy recommendations for heritage tourism development that address implementation challenges while supporting community priorities and cultural sustainability across multiple governmental and organizational levels.

### Municipal and regional government policy framework

Municipal governments should develop comprehensive ICH asset mapping initiatives that understand dynamic interconnections between cultural practices, community relationships, and spatial configurations rather than focusing solely on individual heritage sites or isolated cultural elements. Policy frameworks should explicitly integrate ICH preservation and tourism development with broader urban renewal initiatives through coordinated planning processes that avoid bureaucratic silos and competing agency priorities.

Recognition of ICH temporal rhythms tied to seasonal cycles, agricultural calendars, and traditional observances requires tourism development approaches that respect cultural timing rather than imposing commercial scheduling priorities. Support for hybrid governance model development should formally recognize and empower traditional cultural authorities alongside modern administrative structures through legal frameworks that enable community control over heritage tourism development while ensuring appropriate regulatory oversight and quality standards.

### Tourism industry development guidelines

Tourism developers and operators should shift operational approaches from presenting static heritage displays toward facilitating co-created, immersive cultural experiences that enable authentic participation while respecting cultural boundaries and community priorities. Active visitor participation in culturally appropriate rituals, workshops, and community events should focus on embodied knowledge transmission and collective experience rather than purely cognitive information transfer or passive observation.

Investment in innovative narrative strategies should employ multiple media platforms and multilingual interpretation while maintaining narrative coherence and cultural authenticity through community involvement in content development and presentation. Interpreter training should emphasize deep cultural understanding and respectful engagement rather than standardized factual information delivery or simplified cultural explanations that reduce complex traditions to tourist-friendly narratives.

### Community empowerment and capacity building

Communities and cultural practitioners should receive targeted support for strengthening internal organizational capacity including dragon boat associations, ancestral hall committees, and other traditional institutions that enable effective

engagement with external stakeholders and successful negotiation of tourism development terms that protect community interests while enabling economic benefits.

Active documentation of ICH in diverse formats including oral histories, video recordings, and digital archives should ensure knowledge transmission continuity while informing authentic tourism interpretation that reflects community priorities and cultural values rather than external commercial interests or simplified tourist expectations.

Communities should develop selective engagement strategies that enable strategic decisions about which cultural aspects to share publicly while protecting sacred, private, or vulnerable cultural practices from inappropriate commercialization or misrepresentation. Mechanisms for directly capturing tourism benefits through community-owned enterprises, local employment opportunities, and financial contributions to community-controlled heritage funds should ensure that tourism development generates equitable community benefits rather than external profit extraction.

## Limitations, future research directions, and theoretical development

This study's limitations provide important directions for future research development and theoretical refinement. The temporal scope of two-year intensive fieldwork provides comprehensive contemporary dynamics analysis but cannot assess long-term cultural and economic outcomes of tourism development strategies, requiring longitudinal research tracking heritage tourism impacts across decades to validate whether identified cultural invariants persist through sustained external pressure and whether tourism-supported cultural communities maintain economic and social sustainability over extended periods.

Furthermore, regarding the sampling structure, we acknowledge a disparity between the demographic composition of the village and our respondent pool. Although migrants constitute approximately 80% of Chebei's resident population, our qualitative sample included only 6 migrant tenants and 3 tourists. This limitation stems from the fact that the core rituals and organizational activities of the Dragon Boat Festival remain dominated by local clans with ancestral ties. The consequence of this sampling limitation is that our analysis may disproportionately reflect the perspectives of active cultural insiders and traditional stakeholders, potentially underrepresenting the nuanced, lived experiences of the silent majority—the migrants. While our findings suggest that "cultural experience "acts as a bridge, the specific barriers or passive forms of participation experienced by this large demographic may not be fully captured. Future studies should employ broader quantitative surveys or targeted sampling of non-local residents to rigorously assess the inclusivity of such ICH spaces.

Single case study focus enables rich theoretical development and deep cultural understanding but limits direct empirical generalizability to diverse cultural contexts, though the theoretical framework remains conceptually transferable across different heritage tourism situations. Language and cultural access considerations may limit research insight despite extensive researcher immersion and local collaboration, requiring future research involving diverse cultural backgrounds and linguistic capabilities to capture subtle cultural meanings and community perspectives that may remain inaccessible to external researchers.

Future research should pursue systematic comparative studies applying topological framework analysis to diverse ICH sites globally, testing theoretical generalizability while refining understanding of how specific cultural invariants manifest under different environmental conditions, institutional frameworks, and tourism development approaches. Longitudinal analysis tracking Chebei and similar heritage tourism sites across multiple decades would validate whether cultural invariants genuinely persist through sustained transformation pressures and whether community-controlled tourism development generates lasting economic and social benefits.

Quantitative validation research should develop systematic measures for topological properties within cultural systems, enabling statistical comparison and hypothesis testing that complements qualitative theoretical development. Design research engaging directly with communities and tourism stakeholders in participatory intervention development guided by topological framework principles would refine heritage preservation approaches while generating practical implementation tools for community use.

Investigation of digital dimensions within heritage tourism should explore how emerging technologies create new forms of heritage space topology through virtual tourism experiences, social media cultural transmission, and digital community formation that may transform spatial relationships and cultural practice in ways requiring theoretical development and policy consideration.

## Conclusions

This research pioneers the application of topological framework analysis for understanding complex ICH space dynamics within rapidly urbanizing contexts, offering a systematic approach for heritage tourism management and policy development. Using Chebei Dragon Boat Scenery as an exemplary case study, we demonstrate that cultural practices represent dynamic entities capable of profound adaptation while maintaining essential identity properties and community relationships that define cultural belonging and enable transgenerational transmission.

Our primary theoretical contributions include identifying "cultural experience" as the core ICH topological invariant that provides dynamic authenticity criteria transcending rigid material preservation requirements to encompass deeper relational and embodied meanings that enable cultural continuity through environmental transformation. We develop empirically-derived "culture-power-capital" three-dimensional analytical framework that systematically explains how ICH spaces undergo production and reproduction through dynamic interactions among cultural values, power relationships, and economic forces, with tourism serving as significant catalyst that can either support or undermine cultural sustainability depending on community involvement and management approaches.

We articulate comprehensive "2-6-18" generative logic that captures urban village ICH space remarkable resilience and adaptive capacity while providing systematic analytical structure applicable across diverse heritage tourism contexts. This framework enables understanding of how cultural practices adapt across individual, community, regional, and temporal scales while maintaining essential characteristics that define cultural identity and community belonging.

Practically, this research offers holistic approaches for urban village ICH space revitalization and sustainable utilization that directly inform heritage tourism policy development and community-controlled management strategies. We demonstrate how "urban renewal + cultural tourism integration" can serve as powerful synergistic spatial reproduction mechanism that transforms ICH from vulnerable cultural relics into living, economically viable, culturally enriching community assets that actively drive high-quality urban development while maintaining resident quality of life and cultural authenticity.

By providing concrete implementation strategies for urban planners, tourism managers, and local communities, this study contributes to fostering adaptive sustainability within heritage tourism development that ensures invaluable cultural heritage continues thriving and evolving within rapidly changing global urban environments. The topological framework offers robust analytical tools that transcend traditional preservation-development dichotomies toward comprehensive understanding of how cultural continuity can persist through rather than despite transformation processes.

In our rapidly urbanizing world, the fundamental question is not whether heritage spaces will undergo transformation but how transformation processes can occur while preserving rather than disrupting cultural topology and community relationships. This study provides both conceptual frameworks and practical guidance for navigating these complex transformation processes while honoring both cultural continuity requirements and contemporary urban development needs.

The COVID-19 pandemic and accelerating digital transformation make these analytical considerations increasingly urgent for heritage tourism development. As heritage spaces increasingly span physical and virtual domains, maintaining cultural invariants requires innovative topological thinking that recognizes culture's adaptive capacity while supporting community control over transformation processes. Yet fundamental principles remain consistent: identify and protect essential cultural relationships and experiential qualities while embracing transformative possibilities that enhance rather than undermine community welfare and cultural vitality.

Looking toward future development, accelerating global urbanization makes understanding heritage adaptation mechanisms increasingly critical for human cultural diversity and community wellbeing. As traditional communities

worldwide face development pressures that threaten cultural survival, finding approaches that enable cultural continuity while supporting economic sustainability proves essential for both local community welfare and global cultural heritage preservation.

Tourism, despite its complex and sometimes problematic history with cultural authenticity and community empowerment, can provide viable mechanisms for living heritage valorization and community support when thoughtfully implemented through community-controlled development processes that prioritize cultural priorities and resident welfare over external commercial interests.

The topological framework developed through this study provides conceptual tools and practical strategies for addressing these contemporary challenges while respecting culture's dynamic nature and communities' rights to control their cultural development directions. Future research should systematically test and refine these insights across different cultural contexts, develop practical implementation tools for community use, and track long-term outcomes of topological framework-guided heritage tourism development.

As heritage studies continues evolving beyond static preservation paradigms toward dynamic living heritage approaches that recognize culture's adaptive nature, understanding how cultural communities maintain continuity through transformation becomes essential for both theoretical development and practical heritage management. This study contributes to that evolving understanding while acknowledging substantial work remains to translate theoretical insights into sustainable community-controlled practice that honors both cultural integrity and contemporary development needs.

Dragon boats racing through Chebei's once-polluted but now-restored waters, paddled by local residents and migrants working together, watched by community members and tourists alike, embody heritage's remarkable adaptive capacity and community resilience. Like the mathematical example of a coffee cup topologically deforming into a donut while maintaining essential properties, Chebei Village has undergone dramatic transformation while preserving core cultural characteristics and community relationships that define local identity and enable cultural transmission.

This represents neither perfect preservation nor tragic cultural loss but creative adaptation—culture folding historical memory and ancestral wisdom into contemporary spatial arrangements and unfolding through innovative practices that honor both past inheritance and future possibilities. Understanding and supporting these complex adaptive processes represents both significant theoretical challenge and practical imperative for heritage tourism development within our rapidly transforming global urban landscape.

## Supporting information

**S1 File. Comprehensive interview guide 20250611.**
(DOCX)

**S2 File. Detailed Grounded Theory Coding Structure 20250611.**
(DOCX)

## Author contributions

**Conceptualization:** Xixi Tang.

**Data curation:** Xixi Tang, Shengchao Li.

**Formal analysis:** Xixi Tang, Shengchao Li.

**Funding acquisition:** Xixi Tang.

**Investigation:** Xixi Tang.

**Methodology:** Xixi Tang.

**Project administration:** Xixi Tang.

Resources: Xixi Tang.

Software: Xixi Tang, Shengchao Li.

Supervision: Xixi Tang.

Validation: Xixi Tang, Shengchao Li.

Visualization: Xixi Tang, Shengchao Li.

Writing – original draft: Xixi Tang, Shengchao Li.

Writing – review & editing: Xixi Tang, Shengchao Li.

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
