## [Decision Letter · Decision Letter 0]

1 Oct 2025

Dear Dr. Li,

Thank you for submitting your manuscript to PLOS ONE. After careful consideration, we feel that it has merit but does not fully meet PLOS ONE’s publication criteria as it currently stands. Therefore, we invite you to submit a revised version of the manuscript that addresses the points raised during the review process.

We look forward to receiving your revised manuscript.

Kind regards,

Tianlong You, Ph.D.

Academic Editor

PLOS ONE

2. In the ethics statement in the Methods, you have specified that verbal consent was obtained. Please provide additional details regarding how this consent was documented and witnessed, and state whether this was approved by the IRB.

3. Please note that PLOS One has specific guidelines on code sharing for submissions in which author-generated code underpins the findings in the manuscript. In these cases, we expect all author-generated code to be made available without restrictions upon publication of the work. Please review our guidelines at https://journals.plos.org/plosone/s/materials-and-software-sharing#loc-sharing-code and ensure that your code is shared in a way that follows best practice and facilitates reproducibility and reuse.

4. We note that Figure 2 in your submission contain [map/satellite] images which may be copyrighted. All PLOS content is published under the Creative Commons Attribution License (CC BY 4.0), which means that the manuscript, images, and Supporting Information files will be freely available online, and any third party is permitted to access, download, copy, distribute, and use these materials in any way, even commercially, with proper attribution. For these reasons, we cannot publish previously copyrighted maps or satellite images created using proprietary data, such as Google software (Google Maps, Street View, and Earth). For more information, see our copyright guidelines: http://journals.plos.org/plosone/s/licenses-and-copyright.

1. You may seek permission from the original copyright holder of Figure(s) [#] to publish the content specifically under the CC BY 4.0 license.

Additional Editor Comments (if provided):

Reviewers' comments:

Reviewer's Responses to Questions

**Comments to the Author**

1. Is the manuscript technically sound, and do the data support the conclusions?

Reviewer #1: Partly

Reviewer #2: Partly

Reviewer #3: Yes

2. Has the statistical analysis been performed appropriately and rigorously?

Reviewer #1: Yes

Reviewer #2: Yes

Reviewer #3: Yes

3. Have the authors made all data underlying the findings in their manuscript fully available?

Reviewer #1: Yes

Reviewer #2: Yes

Reviewer #3: Yes

4. Is the manuscript presented in an intelligible fashion and written in standard English?

Reviewer #1: Yes

Reviewer #2: Yes

Reviewer #3: Yes

Reviewer #1: The paper Folding and Unfolding: A Topological Framework for Understanding Intangible Cultural Heritage Tourism in Urban Villages - The Case of Chebei Dragon Boat Scenery, Guangzhou, China takes the Intangible Cultural Heritage (ICH) of Chebei Dragon Boat Scenery in the "urban village" of Guangzhou, Guangdong Province, China as a case study. It obtains reliable data through field observations and interviews, and uses topological theory to launch discussions from the disciplinary perspective of heritage tourism management. The paper appropriately applies methods, has a relatively complete literature review, complies with research ethics, and follows citation norms. In particular, the borrowing of topological theories is a theoretical highlight of this paper.

However, there are still some issues in the current paper that require further consideration, specifically:

First, regarding the theory. The theoretical applicability of using topological theory to analyze intangible inheritance needs further clarification. This is specifically reflected in the following two aspects: (1) ICH inheritance involves multiple dimensions such as power games, economic interests, and social identity. Although the topological "folding-unfolding" model can explain the transformation of cultural meanings, it is difficult to fully cover specific issues in ICH protection such as institutional contradictions (e.g., conflicts between government planning and community autonomy) and economic exploitation (e.g., commercial abuse of ICH symbols by capital). For example, the paper mentions the "conversion of symbolic capital into economic value" of Chebei Dragon Boat, but the topological framework has relatively weak analysis of "how capital distorts cultural meanings". (2) "Invariants" in topology (such as connectivity and dimension) have strict mathematical definitions, while the paper defines "cultural experience" as a "topological invariant", which is essentially a metaphorical borrowing rather than a strict conceptual equivalence. For instance, "invariants" in mathematics are objective properties verifiable through algorithms, whereas "cultural experience" is highly subjective and context-dependent (e.g., differences in understandings of dragon boat rituals among different groups), which may lead to ambiguity in the definition of "invariant". Therefore, further empirical evidence is needed for "cultural experience".

Second, regarding the methods. The relationship between the questions listed in the paper's interview outline and the coding categories is not clearly explained. The author used the method of participant observation in the research process, and the feelings obtained from participant observation must have influenced the construction of coding categories in the paper, which needs to be explained in the paper. Even if it is impossible to supplement the sample of interview subjects, some intuitive feelings obtained from participant observation (usually corroborated by field notes) can be used for supplementary explanation. Otherwise, it is difficult to derive such rich coding categories from the currently presented interview outline.

Reviewer #2: Comments to the manuscript:

As a space of hybrid communities within rapidly urbanizing Chinese cities, the urban village offers a compelling site for examining the viability and continuity of intangible cultural heritage (ICH) under transformative pressures. This manuscript seeks to observe this phenomenon through a novel Topological Framework, applying it to the case of the Chebei Dragon Boat Scenery in Guangzhou. While the study is theoretically innovative and empirically rich, several aspects of its presentation and analytical depth require further clarification and revisions:

1Engagement with ICH literature:

Although the manuscript is thematically centered on ICH, it falls short of fully engaging with international debates and scholarly literature in the field—particularly regarding concepts such as living heritage, community participation, and cultural sustainability. Several central terms—continuity, authenticity, cultural experience, and cultural space—are employed without adequate theoretical grounding or reference to international academic discussions. For instance, how does the paper’s understanding of cultural space differ from or build upon established definitions in cultural geography or UNESCO discourse? A clearer conceptual framework is needed to prevent ambiguity and strengthen theoretical credibility.

2Theoretical framing and literature foundation:

The authors are proposing a topological perspective and a “culture–power–capital” matrix. However, the conceptual framework requires stronger support from relevant English-language literature, particularly in heritage studies, urban cultural geography, and sociology of space. Terms like cultural space and symbolic capital have rich scholarly lineages that should be critically reviewed and situated within existing debates. Without this, the theoretical innovation remains underdeveloped and risks appearing detached from broader academic conversations.

3Research participants:

Given that migrants constitute over 80% of Chebei’s current population, they play a pivotal role in the community's structure and the transmission or transformation of ICH. However, the study includes only six migrant participants and three tourists, while disproportionately focusing on traditional stakeholders. This sampling limits the analysis of hybrid identities and social dynamics within urban village ICH practices. A more balanced and inclusive sampling strategy would better reveal the contested, negotiated, and pluralistic nature of ICH in contemporary urban settings in China.

4Dialogue with international ICH and tourism studies:

The intersection of ICH, tourism, and urban transformation has been extensively discussed in international research. To enhance the global relevance of this study, the authors should engage more critically with the existing international literature on ICH tourism and the effects of urbanization and changing constitution of community. The current analysis tends to emphasize the positive role of tourism and “urban renewal + cultural tourism integration” for ICH, without addressing potential risks such as cultural appropriation, or loss of community agency and control. A more nuanced, dialectical perspective would strengthen the critical depth of the conclusions.

Reviewer #3: Detailed review of the article titled “Folding and Unfolding: A Topological Framework for Understanding Intangible Cultural Heritage Tourism in Urban Villages”

This article is a solid analysis of intangible cultural heritage (ICH) sustainable development based on an empirical case study, that of the Chebei Dragon boat festival, in China. The article’s main objective is to propose a general model for heritage tourism that considers the adaptative dynamics of cultural continuity *in relation to development*. This proposal, which the authors present as a “topological framework”, is *designed for heritage tourism management*.

My emphasis above is meant to underlines the article’s specificity and limits. The presentation of the general framework, in the context of the “theoretical bridging” proposed between topology and cultural heritage studies deserves further discussion and clarification. I will address these two aspects separately in some general remarks below, starting with 1) the dynamic approach in heritage studies, and then turning to 2) the topological model as a way to formalize this dynamic nature. The remaining of this review will address specific questions that follow on some of these general remarks, and offer some suggestions regarding specific claims made in the paper.

General remarks:

1)Regarding cultural heritage studies properly speaking, the idea advanced (p. 7/65) that “ICH's multidimensional conceptual system remains fundamentally stable” should be nuanced. To the contrary, the notion of “safeguarding” recognizes the dynamic nature of cultural heritage, where transmission and reproduction imply the possibility of change. Beside the official wording of the convention itself (https://ich.unesco.org/en/safeguarding-00012) there is a significant literature on these aspects of transmission and evolution, as the continuity of ICH depends also on its (evolving) relevance for the communities, in relation to their identity.

I would therefore challenge the assertion (p. 7/65) that “Traditional approaches often conceptualize heritage spaces as static entities requiring protection from transformation, rather than recognizing the dynamic nature of living heritage that thrives through continuous practice and transmission”. While this may have been true in some times past, there exist many studies that conceptualizes heritage in a non-static way (as is already clear in the notion of “safeguarding”). Therefore, the contrast the authors point out (between static/dynamic) in order to justify the need for a different (topological) approach is somewhat flawed.

Their approach, however, remains valid in itself and interesting, but proper acknowledgement of scholarly achievements through literature review is warranted, and some further discussion would strengthen the article and help better contextualize the authors’ intervention.

In sum, if one was to recognized from the start (as I believe one should) that “heritage” (like “culture” or “tradition”) is not a fixed set of cultural content or properties, but an evolving system for which change is actually a condition of its reproduction, we would be closer to the logic of the topological metaphor, which is not so novel in itself (see next point). In other words, the idea that there can be continuity within change seems to me to be generally accepted in social sciences, and in heritage studies, and should rather be a starting point.

2)Regarding the topological model, the authors present their source of inspiration in mathematical topology in an accessible and appropriate way. The authors’ wish to use a topological model to analyze “cultural persistence within spatial transformation” is certainly relevant, if not new in terms of our understanding of cultural processes.

While some general references are given in relation to their use of topology as a model, the article does not appropriately acknowledge the extensive body of literature in social sciences that mobilize the reference to topology.

Within this extensive literature, I would recommend the article by Lury at al. (DOI: 10.1177/0263276412454552), representative of what some have called a “topological turn”, and ensuing discussions in Martin and Secor regarding geography (DOI: 10.1177/0309132513508209) that are relevant here in regards conceptualization/theorization of space, and important methodological considerations in Phillips (DOI: 10.1177/0263276413480951). (And more specifically in the context of heritage studies, would B. Rudolff (2006), ‘Intangible’ and ‘tangible’ heritage: A topology of culture in contexts of faith, deserve a mention and/or discussion?)

My recommendation is not so much that these authors (among others) should be cited, but that their respective interventions be considered so as to inform, and help refine, the approach proposed here in this article. In fact, what is not clear is if the authors use the reference to topology as a source of specific tools for analysis, that is, as an analogy, or simply as a metaphor.

As the authors claim (p. 8/65) the topological model can “enables identification of invariant properties within ICH practices while their physical manifestations and social contexts undergo significant change.” They advance two components for the relevance of this model: 1) “cultural continuity that transcends material fixity”; 2) metaphorical tools of folding and unfolding (by which cultural meanings are transformed, made implicit/explicit).

I would argue that the first component is a standard understanding of cultural processes that does not require a topological model; and would agree that the second one is indeed only relying on topology as a metaphor (and the “fold”, in its Deleuzian inspiration, is only remotely related to mathematical topology properly speaking).

When the authors claim (p. 13) that they use “inductive theory building”, while this may be true in general regarding their overall proposal, the topological insights, in my view, only loosely emerge from data: “cultural continuity” (which the authors refer to as an “invariant”) paired with “transformation” can only be described as topological at the metaphorical level.

Therefore, at this stage, “topology” is only used to draw attention to and formulate a property that is already recognized in social sciences, that of the “transformational” continuity of culture or tradition. It is not clear that (mathematical) topology is providing any tool here, and the “bridging” (as described here of p. 49) advocated for here lacks any clear and rigorous translation (functional mapping) from one domain (mathematics) to the other (cultural activity).

These two general comments above do not invalidate the proposal made in this article; they call for nuance, better contextualization within the field of social sciences, further clarity on the scope and goals, and detailed explanation of the technical translation of topology to the field of heritage study (if the authors wish to pursue in this direction). My detailed remarks below aim at providing further suggestions regarding how their proposal could be refined, or revised.

In my view, what the article also needs to make very clear throughout, is that the approach they propose applies to, and may be needed more specifically in, the context of the management of ICH. In this specific field (and more precisely that of heritage tourism management), their intervention and proposal for a more dynamic model (i.e., an adaptive development model) is then more justified.

Detailed remarks and further suggestions:

The general question of ICH space dynamics is crucial for the article’s proposal, as it is to this “space” that the topological model applies specifically.

What are the specific qualities of the (ICH) “spaces” considered here? The various questions raised (p. 5/65) regarding ICH spaces’ adaptation, relational logic, etc. in the context of urban spaces, will all require first a clear definition of what “ICH spaces” actually means. The “spatial” component of “ICH spaces” is only explicated on p. 8-9/65. I feel this is a key section of the paper and important component of the conceptualization offered by the authors.

Similarly, in the context of urban development, what is identified as a need for research to explore (p. 6/65) “intangible spatial transformation involving social behavior, spatial relationships, discourse systems, conceptual order” it is not clear in which sense the transformations alluded to (which?) are spatial, and therefore what “intangible spatial transformation” actually is.

Regarding how ICH spaces are being characterized (p. 8-9), with their temporal and spatial dimensions: it is stated that the mechanism of “temporal folding” and the connection point “align remarkably with topology's concept of topological invariance” and that they “constitute specialized terminology within topology”. This should be explained further: the import of a particular terminology does not imply that its application to a new domain is straightforward. While the topological vocabulary is used in a relevant and appropriate way, when “cultural space” is described (p. 10) as having topological properties “through its capacity for folding temporal layers”, the “fold” itself is not enough to explain what these properties are.

Regarding how "cultural experience" is identified as the core topological invariant that enables cultural continuity despite spatial transformation, there appear (to this reader) to be an ambiguity: it seems the authors are talking here about cultural processes in general – while in fact they are really focusing of “culture” in the context of ICH properly speaking (that is, lined to various institutional, economic factors), and more specifically on what they call “ICH spaces”.

Finally, the production of ICH spaces is detailed in the section p. 9/65, in relation to a “culture-power-capital” matrix. This matrix combines cultural spaces, power spaces and economic spaces which produce "ICH space assemblages"—complex configurations where cultural meaning, social power, and economic value interact in mutually constitutive ways.” These “assemblages” are reconfigured by tourism that is a driver for change.

I find this formulation and analysis stimulating, and I feel that (beside the use of vocabulary that aims at keeping the link to general topological theorization, such as “assemblage”) the authors offer a reading and interpretation quite close to theories developed by sociologist Pierre Bourdieu. I would recommend the authors have a close consideration of how their “matrix” echoes key aspects of Bourdieu’s conceptualization of “social space” and various forms of capital (cultural, economic, symbolic…). I suggest the authors consult M. Meissner (2021), Intangible Cultural Heritage and Sustainable Development, which offers a robust transposition of Bourdieu’s theories to the field of heritage study and can be a source of inspiration. If the authors wish to then pursue further with a topological interpretation, I would recommend they explore N. Fogle (2010), The Spatial Logic of Social Struggle: A Bourdieuian Topology, especially the chapter on “Social Topology”.

For example, for the authors the category of “cultural experience as topological invariant” is made of the “experiential qualities” of participants (it does not correspond to physical forms or practices). Couldn’t these experiential qualities be otherwise called tradition? The notion of “tradition” itself is of course tricky, but it has long been recognized that there is an internal dynamic in all traditions by which they appear not static (as the common view holds): as the saying goes, the more it changes the more it is the same. Or, would the Bourdieuian notion of habitus be helpful here, especially given Bourdieu’s emphasis of the body and the notion of practice?

The article describes a “cultural infrastructure” that is “integrated” so that it supports both “traditional community functions” and “tourism development”, therefore enabling maintenance of “cultural authenticity” (P. 15): this constitutes what the authors identify as “the material foundations for the cultural experiences that constitute […] topological invariant”. First, I think it is necessary to emphasize that the infrastructure is the result of significant intervention and urban planning and that the continuity of the cultural experience (which does not equate with “authenticity”) is not necessarily the result of the “maintenance” of tradition, but can very much be the result of this very planning, therefore not a cultural continuity per se, but a targeted use of cultural resources for particular (economic, touristic…) ends. In other words, the very claim of “invariance” in this regard is problematic. For the authors, it shows that “ICH spaces maintain essential properties through […] transformation” linked to tourism development. Are these properties maintained (by which means?) or are they part of the overall design of (tourism) development?

The “generative logic” identified (p. 23 ff) as underlying ICH space dynamic is the result not of properties of the cultural experience per se but of how the authors structured their data. The overall design of the main and sub-categories makes for a fine and effective analytical framework. I do not see the “topological structure” the authors identify in the three main categories (ICH evolution, ICH presentation and ICH experience), however, the analysis is fine and I find this (more descriptive) section of the paper successful and the argumentation convincing. There a many stimulating interpretations, even if at times the topological model finds its limits (depending of if the authors decide to use it beyond the level of metaphor.)

ICH evolution: here the authors mobilize the notion of “folding”, a (topological) process by which evolution is not linear but a layering of symbolic space. They describe a process of “ritual compression” despite surface secularization. As they state (p. 27): “The sacred dimension persists not through unchanged replication but through adaptational continuity that preserves relational meanings within altered forms.” The key term here is “relational meanings”. Similarly, when (p. 28) they describe the maintenance of “water roads” networks: “These persistent social networks demonstrate topological continuity where relational structures endure through radical environmental transformation, maintaining cultural integrity across spatial and temporal scales.” The key term here is “relational structures”. In both cases, however, while it is crucial to recognize the dialectic between change of enduring relations, what is properly topological? Or again (p. 29), about how cultural transmission happens despite partial understanding, in which sense can it be said to demonstrate “topological robustness”?

ICH experience: it is for the authors the core topological invariant maintaining cultural continuity across time, space and social change. Made up of three interrelated elements: embodied knowledge (practice), collective effervescence (<durkheim), as="" authors="" connection.="" state="" temporal="" the="" transgenerational="">Powerspace reproduction: traditional authority structures adapt to contemporary governance requirements, but maintain legitimacy in the community. The authors describe this as “power spaces can transform topologically—changing organizational configuration while preserving essential authority relationships and decision-making processes that protect community interests and cultural priorities.” (p. 39) As one participant puts it “modern structure, traditional spirit”.

Or, for another example (p. 41): “The preserved invariant proves to be collective commitment to cultural practice and community welfare rather than exclusive bloodline continuity, demonstrating topological preservation of essential relationships through transformed membership criteria.” The sentence would make as much sense if the words “invariant” and “topological” were removed and the sentence read: “The collective commitment to cultural practice and community welfare over exclusive bloodline continuity demonstrates that the transformation of membership criteria can help preserve essential relationships.” Again, when it is stated, p. 46: “communities demonstrated remarkable topological flexibility”, it seems that the notion of topology is being stretched here beyond recognition.

All these examples, or again that of COVID-19 “as a topological resilience test” find their limit in that this interpretative approach relies on the topological framework of invariance in spite of transformation, instead of demonstrating the structural logic of intrinsic relations. So, the authors are right in saying that (p. 46): “This pandemic response demonstrates topological resilience—the capacity to maintain essential cultural properties through radical deformation of surface practice modes.” But this could be formulated without reference to topology, which does not afford explanatory value here.

To sum up, the article basically describes and demonstrates that cultural heritage preservation can work effectively. The chief goal of the article is to propose a model for ICH-led tourism development as opposed to theme park model: “This model demonstrates how tourism development can facilitate rather than undermine cultural continuity when properly structured around community priorities and cultural values.” And (p. 49): “The approach recognizes culture as lived practice requiring continuous maintenance rather than periodic performance, ensuring authentic cultural experience while supporting sustainable economic development.”

I find the effort in describing how local communities maintain a form of cultural continuity in spite of transformation valuable: but it should be emphasized that what is being described (and what the topological framework is supposed to apply to) is not just the work of a community, but the combined and planned efforts a several actors and institutions at various scales. The relational logics are therefore not so much intrinsic to the cultural phenomenon studied, but instead very much linked to many external (political, economic, cultural, etc.) factors that have shaped not only the transformations described but also the (conditions for) continuity itself. The topological model finds its limit, as it does not take into account the importance of these external factors in shaping the ICH spaces that afford the possibility of cultural continuity.</durkheim),>

**Do you want your identity to be public for this peer review?** For information about this choice, including consent withdrawal, please see our Privacy Policy

Reviewer #1: No

Reviewer #2: No

Reviewer #3: **Yes:**  Stéphane Gros

---

## [Author Response · Author response to Decision Letter 1]

2 Nov 2025

Response to Reviewers

Dear Dr. You, Academic Editor, and Esteemed Reviewers,

Thank you for your meticulous review and the invaluable, constructive feedback on our manuscript. We have benefited greatly from your insights. In accordance with your recommendations, we have undertaken a careful and rigorous revision of the paper, as detailed in the uploaded file "Revised Manuscript with Track Changes."

Below, we detail our point-by-point responses to the comments from the journal office and all reviewers, referencing the specific changes made in the revised manuscript.

Part 1: Response to Journal Requirements

1. Journal Requirement (Style):

Please ensure your manuscript meets PLOS ONE's style requirements…

Response: We have thoroughly reviewed the PLOS ONE style templates and revised the manuscript, including file naming, formatting, and section structure, to ensure full compliance with the journal's style requirements.

2. Journal Requirement (Ethics Statement):

In the Methods' ethical statement, you have indicated that verbal consent was obtained. Please provide more details on how this consent was documented and witnessed, and state if this was approved by the IRB.

Response: We thank the journal office for this important query. We have significantly strengthened our Ethics Statement. As shown in the "Research Quality and Ethical Considerations" section of our revised manuscript, we have replaced the previous statement with a more detailed one, which is now supported by an official Ethical Review Exemption Certificate (Ref: 20251028) issued by the Academic Committee of Guangdong University of Education (our institution's review board). This certificate confirms the study was classified as "minimal risk research" and explicitly approved the research plan, including the verbal consent procedure.

The revised Ethics Statement in the manuscript now reads: "This study was reviewed and approved by the Academic Committee of Guangdong University of Education (Ethical Review Exemption Certificate, Ref: 20251028). The committee determined that the research, which involved interviews and participant observation with non-vulnerable adult populations, constituted 'minimal risk research' and was exempt from routine supervision. All participants were adults. Prior to interviews, the research purpose, methodology, data usage, privacy protection measures, and participant rights (including unconditional withdrawal) were fully disclosed. Verbal consent was obtained from each participant, as this method was reviewed and approved by the ethics committee for this low-risk study. All interview data have been anonymized to protect participant privacy and ensure confidentiality."

We will also upload a translation of this ethics certificate as a supporting file.

3. Journal Requirement (Code Sharing):

Please note that PLOS One has specific guidelines for sharing code…

Response: We confirm that our study did not generate any custom code or software. The analysis was conducted using standard qualitative data analysis software (NVivo 12). Therefore, this requirement is not applicable to our manuscript.

4. Journal Requirement (Figure 2 Copyright):

We note that Figure 2 in your submission contains map/satellite images that may be copyrighted… We cannot publish previously copyrighted maps or satellite imagery created using proprietary data (e.g. Google software…)

Response: We thank the journal office for this critical reminder. To ensure full compliance with the CC BY 4.0 license, we have completely revised the map figure (now Figure 3 in the revised manuscript). As stated in the new figure caption, the map was redrafted using QGIS software, and all map data is sourced from Natural Earth (public domain). This new figure replaces the previous one and resolves any copyright concerns.

Part 2: Response to Reviewer #1

1.1. Reviewer Comment (Theory Applicability & Limitations):

(1) ICH inheritance involves multiple dimensions… Although the topological "folding-unfolding" model can explain the transformation of cultural meanings, it is difficult to fully cover specific issues… such as institutional contradictions… and economic exploitation… the topological framework has relatively weak analysis of "how capital distorts cultural meanings".

Response: We fully agree with this insightful critique. The topological model's strength is in revealing internal structural logic, not in providing a deep causal analysis of external political-economic forces. To address this, we have made two key revisions in the manuscript:

• First, in Section 1.2 ("Spatial Production Theory and the Culture-Power-Capital Matrix"), we have added a new paragraph explicitly stating the model's limitations. We clarify that for deeper analysis of institutional contradictions or the distorting effects of capital, "a more specialized theoretical toolkit, such as institutional analysis or political economy, would be required."

• Second, in Section 4 ("Discussion"), we have added a new critical analysis that directly addresses the potential negative impacts of capital, including the risks of "cultural flattening and hollowing out," the "weakening of community agency," and the "degenerating from a folk ritual… into a pure tourism attractor."

1.2. Reviewer Comment ("Invariant" as Metaphor / Lack of Evidence):

(2) "Invariants" in topology… have strict mathematical definitions, while the paper defines "cultural experience" as a "topological invariant", which is essentially a metaphorical borrowing… "cultural experience" is highly subjective… Therefore, further empirical evidence is needed…

Response: We thank the reviewer for this crucial point. We have revised the manuscript to clarify our methodological position and strengthen our evidence.

• Clarifying the Metaphor: In Section 1.2 ("Theoretical Bridging"), we now explicitly state that our use of "topological invariant" is a "metaphorical borrowing" and not a strict mathematical equivalence. We redefine it in this socio-cultural context as "the core element that maintains a stable, connective function throughout dynamic transformation—which this paper identifies as 'cultural experience.'"

• Strengthening Evidence: In Section 3.1 ("ICH Experience"), we have added a new concluding paragraph to justify why "cultural experience" acts as this invariant. We use interview data to contrast the perspectives of local residents (1d-9), migrant tenants (1d-6), and tourists (1i-1, 1i-2). This evidence demonstrates that despite different subjective viewpoints, a shared, stable core experience of collective emotion and cultural attraction exists, which functions as the stable "connection point" for the community.

1.3. Reviewer Comment (Methods: Participant Observation):

…the feelings obtained from participant observation must have influenced the construction of coding categories… which needs to be explained…

Response: We fully agree. The role of participant observation was understated. In Section 2 ("Participant Observation"), we have added a new paragraph explaining that this immersive fieldwork (e.g., joining training, rituals, and communal meals) provided a "sensory basis" and "contextual clues" for the coding process, which was essential for capturing tacit cultural logic and informing the development of our coding categories.

1.4. Reviewer Comment (Methods: Interview/Coding Link):

The relationship between the questions listed in the paper's interview outline and the coding categories is not clearly explained… it is difficult to derive such rich coding categories from the currently presented interview outline.

Response: This is an excellent point regarding methodological transparency. To address this, we have added a new table (now Table 1) in Section 2 ("Data Analysis Procedures and Systematic Coding") titled "Example of Correspondence Between Core Interview Questions and Coding Categories." This table visually demonstrates how the core interview questions directly correspond to and generate the rich coding categories presented in the study, making the "method-data-analysis" chain clear and robust. (Please note: This addition required renumbering all subsequent tables throughout the manuscript).

Part 3: Response to Reviewer #2

2.1. Reviewer Comment (Engagement with ICH Literature):

…the manuscript… falls short of fully engaging with international debates and scholarly literature in the field—particularly regarding concepts such as living heritage, community participation, and cultural sustainability… A clearer conceptual framework is needed…

Response: We sincerely thank the reviewer for this feedback. We acknowledge that our engagement with international ICH literature was insufficient. We have made significant additions to Section 1.2 ("Theoretical Bridging") to address this. We now explicitly discuss the core international concepts of "living heritage" and "community participation," referencing UNESCO's 2003 Convention and related international scholarship (e.g., [23-25]). This revision situates our study firmly within this global academic dialogue.

2.2. Reviewer Comment (Theoretical Framing & Literature Foundation):

The authors are proposing a… “culture–power–capital” matrix. However, the conceptual framework requires stronger support from relevant English-language literature… Terms like cultural space and symbolic capital have rich scholarly lineages…

Response: We fully agree. To systematically strengthen our theoretical foundation, we have made significant additions to Section 1.2 ("Spatial Production Theory and the Culture-Power-Capital Matrix"). We now clearly articulate that the "culture-power-capital" matrix is rooted in established theoretical lineages, drawing from Lefebvre [29] (spatial production), Foucault [32] (power/space), and Bourdieu [30, 31] (field/capital). We have added these key citations to connect our framework to the broader academic conversation.

2.3. Reviewer Comment (Research Participants / Sampling):

…migrants constitute over 80% of Chebei’s current population… However, the study includes only six migrant participants… This sampling limits the analysis… A more balanced and inclusive sampling strategy would better reveal the contested, negotiated, and pluralistic nature of ICH…

Response: The reviewer correctly points out this limitation. While we cannot retroactively change the sample, we have revised the manuscript to better leverage the existing data and make our sample structure transparent:

• First, in Section 2 ("In-depth Interviews"), we have added a new paragraph detailing the participant composition and explicitly categorizing them into "Traditional local stakeholders" (N=21, 55.3%) and "External and associated populations" (N=17, 44.7%).

• Second, in our analysis (e.g., Section 3.1, "ICH Experience"), we now more intentionally highlight and contrast the views of migrant tenants (1d-6) and tourists with those of traditional stakeholders to demonstrate the negotiated and pluralistic nature of the ICH space.

2.4. Reviewer Comment (Critical Dialogue & Tourism Risks):

The current analysis tends to emphasize the positive role of tourism… without addressing potential risks such as cultural appropriation, or loss of community agency… A more nuanced, dialectical perspective would strengthen the critical depth…

Response: We fully agree that our original analysis was overly optimistic. As noted in our response to R1.1, we have made a major revision to Section 4 ("Discussion"). We have added a new, substantial paragraph that critically analyzes the "urban renewal + cultural tourism" model, drawing from international research. We now explicitly identify and discuss potential risks, including the "flattening of cultural symbols," the "weakening of community agency," and the "erosion by commercial interests." This adds the necessary critical depth and dialectical perspective that the reviewer rightly called for.

Part 4: Response to Reviewer #3

4.1. Reviewer Comment (Dynamic Heritage Premise):

…the idea… that “Traditional approaches often conceptualize heritage spaces as static entities”… is somewhat flawed… the idea that there can be continuity within change seems to me to be generally accepted in social sciences… and should rather be a starting point.

Response: We fully agree with and thank the reviewer for this fundamental theoretical correction. The reviewer is entirely correct that "continuity within change" is the consensus in modern heritage studies, and our "static vs. dynamic" premise was flawed. We have made two systemic revisions to fix this:

• First, in Section 1.2 ("Theoretical Bridging"), we have rewritten the introduction to start from the accepted consensus that ICH is dynamic ("living heritage"), citing the UNESCO convention (as also noted by R2.1).

• Second, we have deleted the flawed sentence (original P7, lines 133-136) that set up this false dichotomy. We now position our topological framework not as a new idea, but as a novel analytical language and heuristic metaphor to better describe how this accepted dynamic process operates.

4.2. Reviewer Comment (Topological Model as Metaphor / Lack of Dialogue):

…the article does not appropriately acknowledge the extensive body of literature in social sciences that mobilize the reference to topology… (Lury et al., Martin and Secor… Phillips)… it is not clear is if the authors use the reference to topology as… analogy, or simply as a metaphor… what the article also needs to make very clear… is that the approach they propose applies to… the context of the management of ICH.

Response: We sincerely thank the reviewer for these insightful recommendations, which have significantly sharpened the manuscript's focus. We have made the following key revisions based on your suggestions:

• Engaging with "Topological Turn": In Section 1.2 ("Theoretical Bridging"), we now explicitly situate our research within the "topological turn" in social sciences, referencing the scholars you suggested (e.g., Lury et al. [24], Martin & Secor [23], Phillips [25]) to connect our work to this broader academic dialogue.

• Clarifying "Metaphor vs. Tool": In Section 1.2, we now explicitly state that we are using topology primarily as a "heuristic metaphor" and "analytical language" [23-25] to describe structure and relations, not as a strict mathematical tool. This directly addresses the reviewer's ambiguity.

• Focusing on "ICH Management": We fully agree that the framework's primary value is in management. We have sharpened this focus throughout the manuscript. The Conclusion (Section 5) now opens by stating the research offers "a systematic approach for heritage tourism management and policy development," and the Policy Implications (Section 4) section is structured around this practical application. This justifies the need for a model that can identify a stable "invariant" (cultural experience) within a process of managed change.

We believe that these comprehensive revisions, as detailed in the "Revised Manuscript with Track Changes" file, have addressed all the reviewers' concerns. We have significantly strengthened the manuscript's theoretical grounding, methodological transparency, and critical depth. We thank you once again for your invaluable guidance.

Sincerely,

Shengchao Li Corresponding Author Guangdong University of Education

---

## [Decision Letter · Decision Letter 1]

23 Nov 2025

Dear Dr. Li,

Thank you for submitting your manuscript to PLOS ONE. After careful consideration, we feel that it has merit but does not fully meet PLOS ONE’s publication criteria as it currently stands. Therefore, we invite you to submit a revised version of the manuscript that addresses the points raised during the review process.

We look forward to receiving your revised manuscript.

Kind regards,

Tianlong You, Ph.D.

Academic Editor

PLOS ONE

Journal Requirements:

Additional Editor Comments (if provided):

Dear Author,

One of the reviewers recommend the decision of minor revision, while the other recommends to accept this revision. I thus decide to ask for another round of minor revision. I will decide whether to accept the future revision on my own, without seeking assistance from these two reviewers.

This is the only suggestion from that reviewer:

<table border="0"> <tbody> <tr> <td>The authors need to explain clearly the limitation and consequence of the data collection in this research that although migrants constitute 80% of the resident of Chebei, only 6 migrant tenants and 3 tourists have been studied.</td> </tr> </tbody></table>

Reviewers' comments:

Reviewer's Responses to Questions

**Comments to the Author**

Reviewer #2: All comments have been addressed

Reviewer #3: All comments have been addressed

2. Is the manuscript technically sound, and do the data support the conclusions?

Reviewer #2: Yes

Reviewer #3: Yes

3. Has the statistical analysis been performed appropriately and rigorously?

Reviewer #2: Yes

Reviewer #3: Yes

4. Have the authors made all data underlying the findings in their manuscript fully available?

Reviewer #2: Yes

Reviewer #3: Yes

5. Is the manuscript presented in an intelligible fashion and written in standard English?

Reviewer #2: Yes

Reviewer #3: Yes

Reviewer #2: The authors need to explain clearly the limitation and consequence of the data collection in this research that although migrants constitute 80% of the resident of Chebei, only 6 migrant tenants and 3 tourists have been studied.

Reviewer #3: (No Response)

**Do you want your identity to be public for this peer review?** For information about this choice, including consent withdrawal, please see our Privacy Policy

Reviewer #2: No

Reviewer #3: **Yes:**  Stéphane Gros

---

## [Author Response · Author response to Decision Letter 2]

26 Nov 2025

Dear Dr. You (Academic Editor),

We seek to express our sincere gratitude for the opportunity to revise our manuscript one final time. We have made every effort to address the specific concern raised by Reviewer #2 regarding the limitation and consequence of our data collection.

Below is our detailed response and the specific changes made to the manuscript.

Response to Reviewers

Reviewer #2 Comment: "The authors need to explain clearly the limitation and consequence of the data collection in this research that although migrants constitute 80% of the resident of Chebei, only 6 migrant tenants and 3 tourists have been studied."

Response: We fully accept this critical observation. We agree that the discrepancy between the demographic reality (80% migrants) and our qualitative sample size (6 migrants, 3 tourists) creates a specific limitation that must be explicitly acknowledged and analyzed. In the revised "Limitations" section (Lines 1194-1207), we have added a detailed explanation of this issue. We clarified that:

1. The Limitation: The sample is skewed towards local stakeholders because they remain the primary gatekeepers and active practitioners of the ritual.

2. The Consequence: The findings may over-emphasize the "insider" perspective of cultural inheritance, potentially underestimating the challenges or specific marginalization experienced by the migrant majority. We have explicitly stated that this limits the generalizability of our findings regarding the "inclusive" nature of the space and calls for future quantitative validation.

Added Text in Manuscript: Furthermore, regarding the sampling structure, we acknowledge a disparity between the demographic composition of the village and our respondent pool. Although migrants constitute approximately 80% of Chebei's resident population, our qualitative sample included only 6 migrant tenants and 3 tourists. This limitation stems from the fact that the core rituals and organizational activities of the Dragon Boat Festival remain dominated by local clans with ancestral ties. The consequence of this sampling limitation is that our analysis may disproportionately reflect the perspectives of active cultural insiders and traditional stakeholders, potentially underrepresenting the nuanced, lived experiences of the silent majority—the migrants. While our findings suggest that "cultural experience " acts as a bridge, the specific barriers or passive forms of participation experienced by this large demographic may not be fully captured. Future studies should employ broader quantitative surveys or targeted sampling of non-local residents to rigorously assess the inclusivity of such ICH spaces.

We believe this assessment of our study's boundaries significantly enhances the rigor of the paper.

Sincerely, Shengchao Li

---

## [Editor Report · Decision Letter 2]

9 Dec 2025

Folding and Unfolding: A Topological Framework for Understanding Intangible Cultural Heritage Tourism in Urban Villages - The Case of Chebei Dragon Boat Scenery, Guangzhou, China

PONE-D-25-31578R2

Dear Dr. Li,

We’re pleased to inform you that your manuscript has been judged scientifically suitable for publication and will be formally accepted for publication once it meets all outstanding technical requirements.

Kind regards,

Tianlong You, Ph.D.

Academic Editor

PLOS ONE
---

## [Editor Report · Acceptance letter]

PONE-D-25-31578R2

PLOS One

Dear Dr. Li,

I'm pleased to inform you that your manuscript has been deemed suitable for publication in PLOS One. Congratulations! Your manuscript is now being handed over to our production team.

Kind regards,

on behalf of

Professor Tianlong You

Academic Editor

PLOS One